# Re-evaluating cloud chamber constraints on depositional ice growth in cirrus clouds– Part 1: Model description and sensitivity tests

Kara D. Lamb[1], Jerry Y. Harrington[2], Benjamin W. Clouser[3], Elisabeth J. Moyer[3], Laszlo Sarkozy[3], Volker Ebert[4,5], Ottmar Möhler[6], and Harald Saathoff[6]

[1]Department of Earth and Environmental Engineering, Columbia University, New York, NY, USA
[2]Department of Meteorology and Atmospheric Science, The Pennsylvania State University, University Park, PA, USA
[3]Department of the Geophysical Sciences, University of Chicago, Chicago, IL, USA
[4]Physikalisch-Technische Bundesanstalt (PTB), 38116 Braunschweig, Germany
[5]Institute of Physical Chemistry (PCI), University of Heidelberg, 69120 Heidelberg, Germany
[6]Institute of Meteorology and Climate Research, Karlsruhe Institute of Technology, Karlsruhe, Germany

**Correspondence:** Kara Lamb (kl3231@columbia.edu)

**Abstract.** Ice growth from vapor deposition is an important process for the evolution of cirrus clouds, but the physics of depositional ice growth at the low temperatures (<235 K) characteristic of the upper troposphere/lower stratosphere is not well understood. Surface attachment kinetics, generally parameterized as a deposition coefficient $\alpha_D$, control ice crystal habit and also may limit growth rates in certain cases, but significant discrepancies between experimental measurements have not been satisfactorily explained. Experiments on single ice crystals have previously indicated the deposition coefficient is a function of temperature and supersaturation, consistent with growth mechanisms controlled by the crystal's surface characteristics. Here we use observations from cloud chamber experiments in the AIDA Aerosol and Cloud Chamber to evaluate surface kinetic models in realistic cirrus conditions. These experiments have rapidly changing temperature, pressure, and ice supersaturation, such that depositional ice growth may evolve from diffusion-limited to surface kinetics-limited over the course of a single experiment. In part 1, we describe the adaptation of a Lagrangian parcel model with the Diffusion Surface Kinetics Ice Crystal Evolution (DiSKICE) model (Zhang and Harrington, 2014) to the AIDA Chamber experiments. We compare the observed ice water content and saturation ratios to that derived under varying assumptions for ice surface growth mechanisms for experiments simulating ice clouds between 180 and 235 K and pressures between 150 and 300 hPa. We found that both heterogeneous and homogeneous nucleation experiments at higher temperatures (> 205 K) could generally be modeled consistently with either a constant deposition coefficient or with the DiSKICE model assuming growth on isometric crystals via abundant surface dislocations. Lower temperature experiments showed more significant deviations from any depositional growth model, with different ice growth rates for heterogeneous and homogeneous nucleation experiments.

## 1 Introduction

Depositional ice growth from vapor is an important microphysical process for cirrus cloud formation. Cirrus clouds form in the pure ice regime (< 235 K), where water in the vapor phase condenses onto ice crystals initially formed either heterogeneously (requiring an aerosol to act as an ice nuclei, IN) or homogeneously (dependent on the water activity of super-cooled aqueous

aerosol droplets (Koop et al., 2000)). Models of cirrus cloud formation have demonstrated that the value of this deposition coefficient would impact the number density of ice crystals formed through homogeneous nucleation (Lin et al., 2002; Gierens et al., 2003), the steady-state supersaturation within clouds (Zhang and Harrington, 2015), and would also have important

implications for the radiative properties of cirrus clouds. Ice crystal complexity, as observed for small ice crystals in cirrus clouds during recent atmospheric field campaigns, is estimated to contribute -1.12 W/m$^2$ cooling towards the earth's radiative budget (Järvinen et al., 2018). In this study, ice crystal complexity refers to surface distortions that affect single ice crystals, in terms of their surface roughness at different scales, polycrystallinity, and hollowing (Schnaiter et al., 2016). This study investigated the impact of ice crystal complexity on the optical properties of ice, but did not assess its impact on depositional

growth rates, which would have additional implications for the persistence and lifetime of cirrus clouds, as well as their potential to dehydrate air entering the stratosphere (Randel and Jensen, 2013).

The habits and growth rates of ice crystals have long been known to sensitively depend on temperature, pressure, and supersaturation, although the surface effects controlling vapor deposition are complex and challenging to characterize experi-mentally. While the faceted growth of ice at temperatures warmer than 253 K has been explained through the temperature de-

35 pendence of surface ledge formation leading to the dominance of different growth mechanisms (Libbrecht, 2005), the growth mechanisms that control vapor attachment kinetics at temperatures lower than 233 K have not been directly measured and remain relatively unknown (Nelson, 2005).

Ice surface kinetic effects in cirrus clouds have often been parameterized in models as a constant deposition coefficient ($\alpha_D$), independent of supersaturation, temperature, and facet. This parameterization assumes that $\alpha_D \ll 1$ for very inefficient

growth, and $\alpha_D \rightarrow 1$ for very efficient growth (Lin et al., 2002; Gierens et al., 2003). However the increasing complexity of ice crystal habits as a function of size observed in atmospheric cirrus clouds (Schmitt et al., 2016; Magee et al., 2021) indicate a constant deposition coefficient cannot represent ice surface processes during all phases of a crystal's growth cycle. Indeed, the evolution of crystal shapes requires deposition coefficients that are not unity (e.g., Lamb and Scott (1974); Libbrecht (2003)). Recent progress on modeling surface kinetic processes on faceted single crystalline ice has been made with the development of

the Diffusion Surface Kinetics Ice Crystal Evolution (DiSKICE) model, which approximates ice with two semi-axes to model crystal habits, allowing for different ice aspect ratios to be consistently modeled during depositional ice growth (Zhang and Harrington, 2014). The theory can capture the growth of crystals with various aspect ratios using a supersaturation, temperature, and facet-dependent deposition coefficient (Harrington et al., 2019), which may make it amenable to modeling more complex crystal forms (Pokrifka et al., 2020).

Various types of experiments have been used to study surface effects during depositional ice growth. Single crystal experi-ments at warmer temperatures (between 233 K and 273 K) are often performed at low pressures to limit diffusive effects. Such studies have indicated the deposition coefficient is a complicated function that depends on the surface supersaturation, the temperature, the presence of chemical impurities, and the structure of the crystal surface (Nelson and Knight, 1998; Libbrecht and Rickerby, 2013; Harrison et al., 2016). However, extrapolating from single crystal experiments performed in controlled

conditions to atmospheric cirrus is not straightforward, as ice crystals growing in the atmosphere compete for available vapor and experience significant changes in ambient conditions due to dynamic processes, such as the influence of gravity waves

(Spichtinger and Krämer, 2013; Jensen et al., 2016; Dinh et al., 2016). These processes will affect the thermal and water vapor environment of the growing crystals, and therefore may influence the manner in which the crystals grow.

Additionally, only a few studies have specifically targeted ice growth at temperatures colder than 233 K (Pratte et al., 2006; Magee et al., 2006; Bailey and Hallett, 2009, 2012) and, to date, no studies have been done on the growth of individual ice facets at these temperatures. Atmospheric measurements have also been used to constrain the deposition coefficient, although there are significant challenges in measuring ice crystal growth rates from an airborne platform, as the evolution of crystal growth over time cannot be directly measured (Kaufmann et al., 2018; Krämer et al., 2020).

One previous study (Skrotzki et al., 2013) focused on measurements of the deposition coefficient at cirrus-relevant temperatures (190-230 K) inside of a cloud chamber, where the evolution of an air parcel can be monitored over time. Cloud chambers simulate cirrus clouds formed in the atmosphere via adiabatic expansion experiments, by rapidly changing temperature and pressure to create ice supersaturated conditions. While single particle experiments have indicated the deposition coefficient can in some cases by very small (Magee et al., 2006) these cloud chamber experiments suggested deposition coefficients in cirrus conditions are near 1, and therefore can generally be neglected in model calculations of mass growth. The past study in AIDA focused on whether surface kinetic effects limit ice growth rates in cirrus conditions, effectively investigating models for the deposition coefficient that assume it is a single constant value, rather than a supersaturation, temperature, and facet dependent function. This constant value for the deposition coefficient can be thought of as the high-supersaturation limit of the deposition coefficient function. Calculating the shape evolution of faceted crystals, even complex ones, would still require estimating parameters for the temperature, supersaturation, and facet-dependent deposition coefficient function, even if this deposition coefficient function does not significantly limit growth rates. Several attempts have been made to explain the discrepancies between different experimental measurements of the deposition coefficient (Libbrecht, 2004; Harrison et al., 2016), but the questions of how these experiments can be reconciled, and whether cirrus clouds form in conditions where surface effects significantly limit growth rates, still remain.

To address these questions, we use observations from expansion experiments performed inside of the AIDA Aerosol and Cloud chamber during the IsoCloud (Isotopic Fractionation in Clouds) campaign (Lamb et al., 2017; Clouser et al., 2020). We investigate whether depositional ice growth models including surface kinetic processes that vary with changes in ambient conditions are consistent with the observed ice growth rates in AIDA. These experiments included cases of both homogeneous and heterogeneous ice formation to investigate the role of ice nucleation pathways on depositional growth. While Skrotzki et al. (2013) explored the variability in the deposition coefficient derived from experiments performed in AIDA at various temperatures and with two types of heterogeneous ice nuclei, the analysis method implicitly assumed that $\alpha_D$ remained constant during a single expansion experiment, investigating only the constant parameterization typically used in cloud models. Recent advances in modeling surface kinetic processes on faceted ice in cirrus clouds (Zhang and Harrington, 2014, 2015) and in experimental measurements of individual ice crystals (Pokrifka et al., 2020) provide motivation to reanalyze these experiments. In this paper (Part 1) we describe the adaptation of a parcel model including different models for surface kinetic effects to the AIDA experiments (Zhang and Harrington, 2014, 2015), and compare predicted ice growth rates under varying assumptions for surface kinetic effects. In a companion paper (Part 2) we plan to quantitatively evaluate constraints placed on surface kinetic

models at low temperatures by these experiments using Bayesian parameter estimation (as in e.g., Schrom et al. (2021)). Here, we discuss models for depositional ice growth from vapor and how the effects of different ice surfaces are parameterized in Section 2. We then discuss the experimental protocol and experiments performed in AIDA in Section 3. In Section 4, we discuss average observed ice growth rates and describe the adaptation of a parcel model with the different models for depositional ice growth to AIDA. In Section 5, we compare observed ice growth rates with predicted ice growth under different assumptions for surface kinetic effects. Finally we discuss the implications of these results for atmospheric cirrus cloud models in Section 6. Since a number of acronyms are used throughout the text, we provide a reference list in Table A1.

## 2 Models for depositional ice growth in cirrus conditions

In single crystalline form, atmospheric ice nucleates and then subsequently grows through vapor deposition as hexagonal ice ($I_h$), with 2 basal (hexagonal) faces and 6 prism (rectangular) faces. The simplest model for depositional ice growth is a capacitance model, which determines mass transfer based on differences between surface temperature and far-field supersaturation (Fukuta and Walter, 1970; Pruppacher et al., 1998). The single crystal growth rate $\frac{dm}{dt}$ in this model is

$$\frac{dm}{dt} = -4\pi D_v f_v A_i C(c,a)(1-S_i)\rho_{v,sat}^{\infty},\tag{1}$$

where $D_v$ is the molecular diffusivity of water vapor in air, $A_i$ is a transfer coefficient incorporating the effects of heat and vapor transport, $f_v$ is the ventilation coefficient, $C(a,c)$ is the capacitance (which depends on $a$ and $c$, half the longest distance along the basal and prism facets, respectively), $S_i$ is vapor supersaturation over ice, and $\rho_{v,sat}^{\infty}$ represents the water vapor density at saturation over ice at the far field temperature (i.e. far from the growing ice crystal) (Mason, 1971). $A_i$ is a function of both the thermal conductivity $k_T$ and the vapor diffusivity $D_v$. In the classical capacitance model, perfect attachment kinetics are implicitly assumed. A consequence of this assumption is that the vapor density is constant over the crystal surface. This surface condition is inconsistent with the growth of any kind of facet, since faceting requires $\alpha_D$ that are less than unity and a vapor density that varies across the crystal facet (Nelson and Baker, 1996).

Ice growth requires consideration of mass and thermal energy transfer through the background gas along with molecular attachment onto the crystal surface. Small ice crystals characteristic of those that form immediately following homogeneous nucleation ($<20\ \mu$m) in cirrus clouds will also be influenced to some extent by attachment kinetics, especially as facets emerge during growth (Harrington and Pokrifka, 2021). The capacitance model can be modified to include these attachment kinetics by assuming growth occurs across two regions: a continuum region, where gases can be modeled as a continuous fluid and diffusion theory applies, and a kinetic region at the surface of the crystal where gases act as their individual component molecules and surface uptake reduces growth (Pruppacher et al., 1998). The boundary between the two regions is the vapor jump length $\Delta_\rho$, which is generally assumed to be the mean free path $\lambda_w$ of a water molecule in air. The surface effects are included as a modified diffusivity $D_w^*$ and a modified thermal conductivity $k_T^*$, which are functions of the deposition coefficient $\alpha_D$ and the thermal accommodation coefficient $\alpha_T$. This latter quantity is thought to be near unity, but it has not been measured at low temperatures.

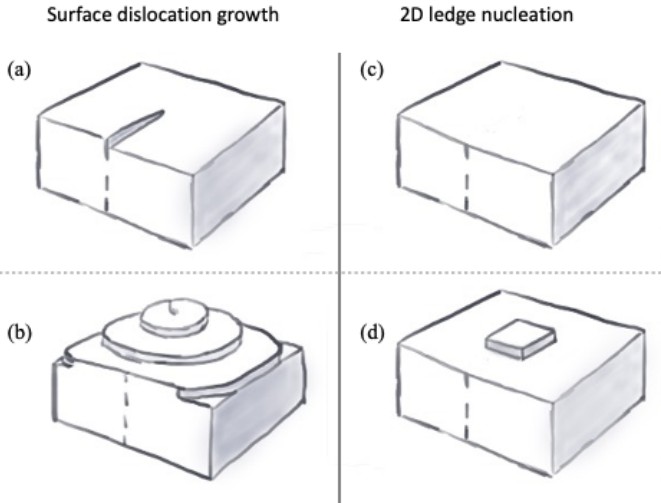

**Figure 1. Examples of crystal surface structure.** One-dimensional defects in the crystal lattice (a) provide favorable attachment sites that lead to screw dislocations (b). When no defects are initially present (c), 2D ledges must first form on the crystal surface to provide favorable attachment sites (d).

When surface ledge densities are high enough that they do not impede growth, the deposition coefficient becomes equivalent to a molecular adsorption/desorption efficiency, e.g., a molecular "sticking" coefficient, where

$$\alpha_D = \alpha_s \tag{2}$$

and $\alpha_s$ is the probability that an incident molecule will adsorb to the crystal surface, i.e. $\alpha_s$ is a constant between 0 and 1. This adsorption efficiency is generally considered to be near unity ($\alpha_s \sim 1$) (Libbrecht, 2005). Experiments using a molecular beam indicated that this sticking coefficient $\alpha_s \sim 1$ and is temperature-independent (Brown et al., 1996).

The constant parameterization for the deposition coefficient (Eq. 2) does not physically account for the diffusion of molecules across the face of the crystal to favorable attachment sites, however. Statistical mechanical considerations indicate that crystal growth through vapor deposition is a function of the structure of the crystal surface, as attachment of vapor molecules to surfaces without pre-existing ledges is not energetically favorable. Ice crystals often form with one-dimensional defects in the crystal lattice leading to the occurrence of screw dislocations (Figure 1a,b). The occurrence of these defects provide attachment sites for vapor depositing on ice, leading to characteristic spiral growth patterns (Burton et al., 1951). Facets without ledges require the nucleation of 2-dimensional "islands", or ledges to provide favorable attachment sites (Figure 1c,d). The deposition coefficient associated with both growth mechanisms has been shown to be dependent on the ratio of the local supersaturation $s_{local}$ to a critical supersaturation $s_{crit}$, with very inefficient vapor deposition associated with 2D ledge nucleation when $s_{local} < s_{crit}$, while screw dislocation growth has a much more gradual dependence (Burton et al., 1951). These growth mechanisms have been observed experimentally, e.g., through advanced optical microscopy of basal and prism facets of ice grown

**Figure 2. The deposition coefficient as a function of surface supersaturation.** Crystals growing via vapor deposition may support different types of growth mechanisms resulting from properties of the crystal surface (e.g., screw dislocations, stacking faults, or 2D ledge nucleation). The deposition coefficient is therefore expected to be a function of the surface supersaturation ($s_{local}$) relative to a critical supersaturation ($s_{crit}$). This critical supersaturation is generally considered to be a function of the crystal facet (basal or prism) and the temperature.

near the melting point (See for example, Sazaki et al. (2010, 2014) and references therein). Other types of crystal defects, such as stacking faults (planar crystal defects), can also act as favorable attachment sites (Nelson and Baker, 1996).

The deposition coefficient for the growth processes described above can be parameterized using a relatively simple model proposed by Nelson and Baker (1996). This model parameterizes the surface diffusion of molecules to ledges or steps that 145 form on the crystal surfaces (Nelson and Baker, 1996), and is an explicit function of the surface supersaturation and properties of the crystal surface, and an implicit function of temperature through the critical supersaturation, $s_{crit}(T)$. Vapor attachment kinetics include both the diffusion time scale across the crystal surface to favorable attachment sites and the relative probability that a molecule will be incorporated into the crystal lattice. Deposition is often parameterized in this case as,

$$\alpha_D = \alpha_s \left( \frac{s_{local}}{s_{crit}} \right)^m tanh \left[ \left( \frac{s_{crit}}{s_{local}} \right)^m \right] \tag{3}$$

where $s_{local}$ is the supersaturation just above the crystal surface (Nelson and Baker, 1996). The adsorption efficiency $\alpha_s$ is generally taken to be unity (see above). Growth mechanisms on ice surfaces are parameterized using different values for $m$ (See Figure 2). Growth by permanent surface dislocation (Burton et al., 1951) is parameterized by a value of $m = 1$, stacking fault-induced nucleation as $m \sim 3-5$ (Ming et al., 1988), and 2D ledge nucleation as $m \sim 10-15$ (Nelson and Baker, 1996). In

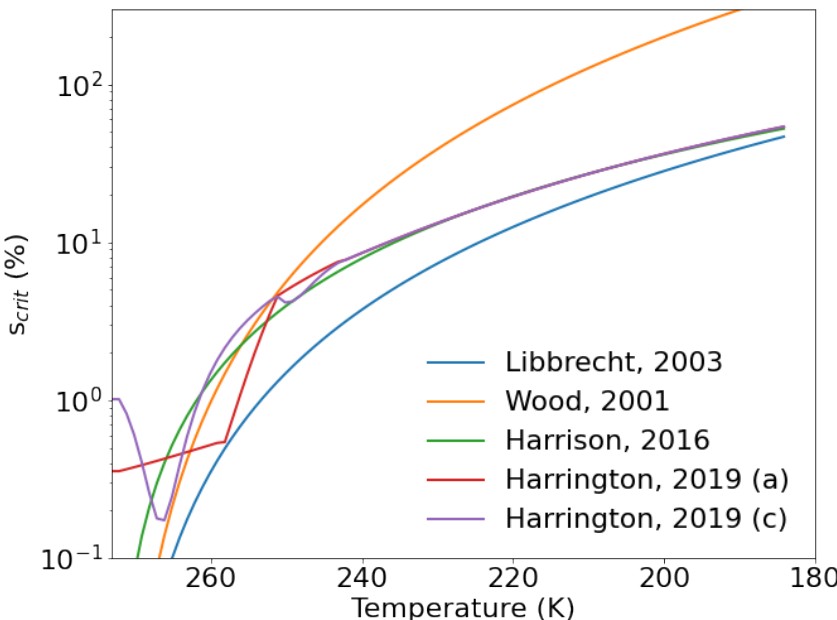

**Figure 3. Temperature dependence of $s_{crit}$.** Parameterizations for the temperature dependence of $s_{crit}$ that we compare in this study.

cloud models and global climate models, even when the sensitivity with respect to the deposition coefficient has been explored, it has generally been varied as a constant value (Eq. 2) and not consistently with the ice surface growth mechanisms (Eq. 3) (Nelson, 2005).

This deposition coefficient is valid for the faceted growth of ice crystals into the variety of habits observed in the atmosphere (Burton et al., 1951; Lewis, 1974; Lamb and Chen, 1995). For $I_h$, the basal and prism facets have been shown to have different deposition coefficients, with distinct dependencies on temperature and supersaturation, consistent with the large variety of ice crystal morphologies observed in the atmosphere. Once deposition coefficients reach unity, morphological instabilities can occur leading to the hollowing of facets and the development of branching arms (e.g., Gonda and Gomi, 1985; Yokoyama and Kuroda, 1990; Wood et al., 2001; Libbrecht, 2005). Recent single crystal experiments have generally focused on the determination of $s_{crit}$ for the basal and prism facets (Nelson and Knight, 1998; Libbrecht and Rickerby, 2013; Harrison et al., 2016), although all of these experiments have focused on measurements at $T >$233 K. Figure 3 shows parameterizations for $s_{crit}$ as a function of temperature. The values below 233 K are extrapolations (Wood et al., 2001; Libbrecht, 2003; Harrison et al., 2016) and rough estimates from prior data (Harrington et al., 2019). Harrington et al. (2019) gives a more full review of past experimental constraints on $s_{crit}$ and their relevant temperature ranges.

Additional complications for ice deposition arise due to the presence of chemical impurities or coatings on ice surfaces (Libbrecht, 2005). Molecular beam experiments have indicated near unity accommodation of water vapor on nitric acid and

acetic acid coated ice, but more efficient desorption in the case of short-chain alcohols (Kong et al., 2014). Previous studies have also demonstrated differences in ice crystal morphology due to the presence of chemical impurities (Hallett and Mason, 1958; Mason, 1971).

## 2.1 Impacts of nucleation pathways on depositional growth rates

Recent experimental work suggests that depositional ice growth rates (Bailey and Hallett, 2002; Harrison et al., 2016; Pokrifka et al., 2020; Harrington and Pokrifka, 2021) and surface complexity (Schnaiter et al., 2016; Voigtländer et al., 2018) may be impacted by whether heterogeneous or homogeneous nucleation initiates the ice growth process. Experiments on heterogeneous ice nuclei have demonstrated surface defects on mineral dust serve as active sites for ice growth (Kiselev et al., 2016). Differences in the morphology of mineral dust has also been shown to impact the number of active sites available for nucleation (Hiranuma et al., 2014a, b). Homogeneous nucleation instead is dependent on a phase transition leading to crystallization in supercooled aqueous droplets, which occurs spontaneously without requiring preferential nucleation sites. Following nucleation, these different processes may lead to ice crystals with distinct surface characteristics. Ice formed as a result of heterogeneous nucleation has been suggested to be dominated by dislocation growth (Harrison et al., 2016), whereas measurements also suggest that homogeneously nucleated ice is initially dominated by dislocations but may slowly transition to ledge nucleation growth (Pokrifka et al., 2020).

Because higher supersaturations are required for homogeneous nucleation, ice crystals formed via heterogeneous or homogeneous nucleation mechanisms experience different histories of supersaturation. This higher supersaturation required to nucleate ice homogeneously could lead to a "kinetic roughening" of the ice surface (Libbrecht, 2005). Additionally, recent experiments of ice formed through homogeneous nucleation within an electrodynamic levitation diffusion chamber at temperatures between 229 and 243 K indicated that a growth transition may occur during growth in conditions of constant supersaturation, with initial growth occurring efficiently and later growth inefficiently (Pokrifka et al., 2020). Ice crystal complexity, defined as the sub-micron surface structure of ice, varies as the crystals grow, with differences between heterogeneous and homogeneous nucleation observed during adiabatic expansion experiments in AIDA (Schnaiter et al., 2016; Järvinen et al., 2018). Homogeneous nucleation experiments were found to create highly complex ice, whereas heterogeneous nucleation experiments lead to ice varying from pristine (molecularly smooth) to highly complex as a function of the supersaturation. Because the deposition coefficient depends on the relative smoothness of the ice crystal surface, these results suggest homogeneous and heterogeneous nucleation may create ice with unique deposition coefficients.

## 3 Cloud chamber experiments of depositional ice growth for T<235 K

To investigate depositional ice growth at low temperatures, cirrus cloud simulation experiments were performed in the AIDA Aerosol and Cloud Chamber during the IsoCloud 4 campaign in March 2013. The IsoCloud experiments consisted of a series of pseudo-adiabatic expansion experiments between 180-235 K, and the experimental protocol used during the campaigns has been previously described in Lamb et al. (2017) and Clouser et al. (2020). Previous studies using the IsoCloud observations

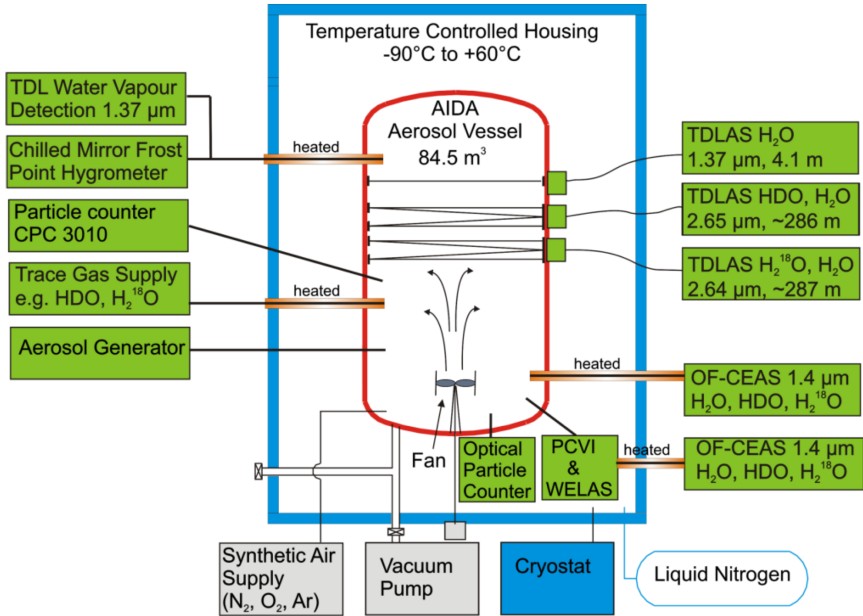

**Figure 4. AIDA measurement chamber and instrumentation during the IsoCloud Campaigns.** The AIDA chamber is a large volume (84.5 m$^3$) environmental chamber. Pressure can be dynamically controlled via vacuum pumps evacuating gases and particles, leading to adiabatic cooling of the gas. A fan circulates air inside the chamber to mitigate gravitational settling and mix the air inside the chamber. The instrumental layout during the IsoCloud experiments is shown, with the TDL instruments (ChiWIS and SP-APicT) measuring water vapor approximately 2/3rds of the height of the chamber, and the welas optical particle counters measuring near the bottom of the chamber. Total water was monitored (by APeT) via extraction along a heated inlet at approximately the same height as the water vapor instruments.

focused on characterizing isotopic fractionation between HDO and H$_2$O during depositional ice growth (Lamb et al., 2017) and investigated saturation vapor pressure over ice for temperatures between 189 and 235 K (Clouser et al., 2020). AIDA is a large chamber (84.5 m$^3$ volume) where mixed phase and ice clouds can be experimentally simulated through adiabatic expansion

of air (a synthetic mixture of N$_2$, O$_2$, and Ar) and water vapor. This adiabatic expansion cools the gas inside the chamber by several degrees Kelvin, leading to conditions of ice supersaturation sufficient to nucleate ice, either by introducing solid aerosol particles (to nucleate ice heterogeneously) or aqueous aerosol droplets (to nucleate ice homogeneously).

In a typical expansion experiment, the cloud chamber is prepared by initially cooling the entire sample volume to a starting temperature < 235 K. This cooling takes several hours and therefore each experimental day focused on a series of 4-7 experi-

210 ments performed with similar starting temperatures. The IsoCloud campaign included experiments over 9 days, for a total of 48 expansion experiments.

To prepare the chamber walls with a thin layer of ice, water vapor was initially sprayed into the chamber and then the chamber was pumped down, with this process repeated several times to provide a reasonably uniform coverage of the walls with a thin ice coating. The ice on the walls helps maintain equilibrium conditions with the water in the vapor phase (with

**Table 1. Optical instruments used in analysis.** The range for Chi-WIS, SP-APicT, APeT indicate the temperature range over which the instrument can generally be used to retrieve water vapor mixing ratios. For APeT, ice crystals greater than 7 $\mu$m are sampled with less than 100% efficiency. The range for the welas instrument indicates the effective spherical radius of the particles that can sampled, although this is dependent on morphology and orientation of the crystals in the instrument.

| Instrument | Observable | Range | Time Res. | Acc. | Description |
|---|---|---|---|---|---|
| Chi-WIS | $H_2O$, *in situ* vapor (ppmv) | 190 - 235 K | 1 s | ±5% | TDLAS at 2.65 $\mu$m, 256-286 m optical path (Sarkozy et al., 2020) |
| SP-APicT | $H_2O$, *in situ* vapor (ppmv) | 205 - 235 K | 1 s | ±5% | TDLAS at 1.37 $\mu$m, 4.1 m optical path (Skrotzki, 2012) |
| APeT | $H_2O$, extractive total water (ppmv) | 190 - 235 K | 1 s | ±5% | TDLAS at 1.37 $\mu$m, 30.3 m optical path (Lauer, 2007; Skrotzki, 2012) |
| welas 1 | Ice crystal number density (#/cm$^3$) | 0.3-46 $\mu$m | 5 s | ±20% | Optical particle counter, PALAS 2000 (Benz et al., 2005) |

the gas initially at a slightly higher temperature than the chamber walls, producing ice sub-saturated conditions with RH$_i$ ∼80-90%).

Ice cloud experiments were performed with several different types of aerosols to create a range of conditions to investigate ice growth inhibition at low temperatures. Before the start of an expansion experiment, aerosols were introduced into the chamber. Arizona Test Dust (ATD), a mineral dust proxy, was used in 31 experiments, as it has previously been shown to be

a very efficient heterogenous IN, with an ice active fraction between 0.6-0.8 for S$_i$<1.15 at T=209 and 223 K (Möhler et al., 2006). ATD particles were size selected via impaction for sizes < 2 $\mu$m and were dispersed via a rotating brush generator (RGB 1000, PALAS) (Möhler et al., 2006). Aqueous sulfuric acid particles ($H_2SO_4/H_2O$, referred to hereafter as SA) were generated as in Wagner et al. (2008) and were used in 9 experiments. Secondary organic aerosol (SOA) was used for 1 experiment with an additional 3 experiments using SOA coated with nitric acid. Atmospheric observations and laboratory measurements have

previously shown that nitric acid may impede ice growth in cirrus conditions (Gao et al., 2004, 2015). Reference pumpdowns (e.g., without any aerosols added) were performed in 6 cases; the typical aerosol background concentrations in AIDA is less than 0.1 cm$^{-3}$. Table 2 gives an overview of all of the experiments performed during the campaign.

In the AIDA chamber, the effective vertical motion (assuming dry adiabatic ascent) can be estimated from the temperature derivative as

$$w_{\text{eff}} = -\frac{dT}{dt}\frac{c_p}{g} \qquad (4)$$

where $w_{\text{eff}}$ is the effective updraft speed in m/s, $\frac{dT}{dt}$ is the time derivative of the temperature in K/s, $c_p$ is the specific heat capacity of dry air (1004 J/(kg K)), and $g$ is the gravitational constant (9.81 m/s$^2$). Typical effective updraft speeds in AIDA during the IsoCloud experiments were between 90-300 cm/s (See Table 2).

After ice formation, the ice on the walls provide an additional source of water vapor to the chamber. In a typical experiment,

pumping occurs over 5-10 minutes, and after the end of pumping, the ice evaporates after ∼10 minutes, as a heat flux through the chamber walls warms the gas inside. A mixing fan located at the bottom of the chamber mediates the impact of gravitational settling and decreases inhomogeneity in the gas. The mixing time scale is on the order of 30 s. The gases and aerosols introduced into the chamber are diluted by a factor of ∼2 while the pumps are turned on. Gas temperature is measured by 5 thermocouples

located along a wire at different heights approximately 1 m from the chamber walls; here we use an average of the 4 lower thermocouples, as the 5th is located near the top of the chamber and shows significantly greater temperature variability; the TDL instruments were located between the 3rd and 4th thermocouples (see Figure 4). The typical standard deviation is <0.5 K in steady-state conditions and < 1 K during the expansion experiments (at 210 K).

Instrumentation used during the campaign provided evolving aerosol and ice number concentrations, total water and water vapor mass mixing ratios, pressure, and temperature during the experiments. A schematic of the experimental set-up during the campaign is shown in Figure 4, and a summary of the instruments used in the experiments, the quantities that they measure, and their associated measurement ranges and uncertainties is given in Table 1. Water vapor was measured via tunable diode laser absorption spectroscopy (TDLAS) sampling directly within the chamber volume. SP-APicT (single pass AIDA PCI in cloud TDL) (Skrotzki et al., 2013) was used for water vapor measurements in dense clouds at warm temperatures (21 of 48 experiments), as its single pass configuration reduces attenuation due to back-scattering from ice crystals. ChiWIS (the Chicago Water Isotope Spectrometer) (Sarkozy et al., 2020) was used to measure water vapor at low temperatures, as its longer path length gives better precision below 205 K. APeT (AIDA PCI extractive TDL) (Lauer, 2007) measured total water (vapor plus ice) via extraction through a heated inlet. Water vapor instruments used in this campaign were tested during the AQUAVIT-II water vapor instrument inter-comparison campaign, and were shown to agree within $\pm2.5\%$ percent (Sarkozy et al., 2020). SP-APicT and APicT were also previously evaluated during the AQUAVIT-I campaign (Fahey et al., 2014).

Uncertainties in mass mixing ratio for TDLAS measurements are generally a combination of effects due to the intrinsic precision of the instrument and other observables, and systematic, multiplicative offsets due to the molecular spectral line parameters. The ChiWIS instrument's typical precision of 22 ppbv in $H_2O$ corresponds to relative precisions of 5% at 0.45 ppmv and 0.02% at 100 ppmv for 1 Hz measurements. In the IsoCloud campaigns, systematic errors due to uncertainties in spectral line parameters are about 3% across all experiments, and systematic biases and uncertanties related to the instrument and its setup contribute another 1.3%. Both APeT and SP-APicT use the same spectral lines to determine water mixing ratios. For consistency, to account for differences due to spectral line strength errors, APeT and SP-APicT were scaled up by 1.5% to match ChiWIS (Clouser et al., 2020; Sarkozy et al., 2020). To determine ice water (subtracting total water from APeT by water vapor from either SP-APicT or ChiWIS), any additional offsets before the start of pumping were subtracted so that initial ice water was assumed to be 0. Additional information about the optical measurements made in AIDA are given in (Wagner et al., 2009). Data for each expansion experiment was synchronized to the start of pumping. ApeT was assumed to have a 17 s offset time (as was previously measured in Skrotzki et al. (2013) by looking at time differences between when *in situ* and extractive hygrometers measured a sudden change in water vapor inside AIDA).

## 4   Modeling depositional ice growth in AIDA

In clouds, depositional ice growth can occur in three distinct regimes, depending on the growth rate limiting processes: vapor diffusion limited, surface diffusion (e.g., kinetically) limited, or heat conduction limited (Kuroda and Lacmann, 1982; Nelson and Baker, 1996). Generally heat conduction limited growth is assumed to impact warm mixed phase clouds, vapor diffusion

**Table 2. IsoCloud 4 Experiments.** Summary of conditions during expansion experiments. Experiments 12, 18, 28, 34, 39, and 44 were reference pumpdowns (with no aerosols introduced into the chamber) and are not shown.

| Exp. | Aerosol | $T_0$ (K) | $\Delta T$ (K) | $p_0$ (hPa) | $\Delta p$ (hPa) | $w_{eff}$ (cm/s) | $r_{v,0}$ (ppmv) | $\Delta r_{v,0}$ (%) | $S_i$ (max.) | CCN (max.) | $R$ ($\mu$m) | Kn |
|---|---|---|---|---|---|---|---|---|---|---|---|---|
| 1 | ATD | 234 | 7.8 | 299 | 65 | -370 | 380 | 39 | 1.21 | 20.7 | | |
| 2 | ATD | 233 | 6.5 | 300 | 100 | -130 | 366 | 17.7 | 1.24 | 8.7 | 5.2-10.4 | 0.04-0.17 |
| 3 | ATD | 233 | 6.4 | 300 | 101 | -120 | 377 | 28.6 | 1.03 | 39.4 | 4.1-6.4 | 0.06-0.12 |
| 4 | ATD | 233 | 9.1 | 300 | 131 | -130 | 375 | 38.2 | 1.21 | 44.0 | 4.4-6.3 | 0.05-0.64 |
| 5 | ATD | 233 | 9.1 | 300 | 132 | -180 | 387 | 43.8 | 1.05 | 51.9 | 5.1-6.3 | 0.06-0.23 |
| 6 | ATD | 223 | 6.6 | 300 | 71 | -170 | 113 | 29 | 1.27 | 11.1 | 1.6-6.8 | 0.05-0.17 |
| 7 | ATD | 223 | 6.4 | 234 | 64 | -140 | 147 | 35.3 | 1.03 | 93.2 | 2.8-3.8 | 0.10-0.13 |
| 8 | ATD | 223 | 8.7 | 300 | 131 | -200 | 114 | 46.4 | 1.04 | 75.0 | 3.7-4.3 | 0.07-0.11 |
| 9 | ATD | 223 | 6.0 | 300 | 71 | -160 | 114 | 30.7 | 1.12 | 70.4 | 3.0-4.5 | 0.08-0.43 |
| 10 | ATD | 223 | 5.5 | 231 | 62 | -130 | 147 | 29.7 | 1.1 | 75.4 | 2.9-4.3 | 0.11-0.61 |
| 11 | ATD | 223 | 8.9 | 300 | 150 | -180 | 115 | 47.3 | 1.03 | | | |
| 13 | ATD | 213 | 5.3 | 234 | 64 | -130 | 40.6 | 33.1 | 1.06 | 351.2 | 1.0-1.6 | 0.27-0.83 |
| 14 | ATD | 213 | 8.4 | 300 | 137 | -160 | 30.9 | 46.8 | 1.04 | 489.9 | 1.0-1.6 | 0.11-0.61 |
| 15 | ATD | 213 | 5.6 | 300 | 71 | -160 | 31.1 | 63.4 | 1.04 | 403.2 | 1.0-1.5 | 0.17-0.56 |
| 16 | ATD | 213 | 5.4 | 234 | 64 | -140 | 39.9 | 32.1 | 1.03 | 455.3 | 1.0-1.5 | 0.30-1.39 |
| 17 | ATD | 213 | 8.4 | 300 | 130 | -150 | 31.1 | 48.3 | 1.04 | 592.1 | 1.0-1.5 | 0.24-1.81 |
| 19 | ATD | 194 | 5.2 | 300 | 71 | -120 | 1.78 | 2.2 | 1.87 | 4.3 | 1.5-2.0 | 0.13-0.19 |
| 20 | ATD | 194 | 4.8 | 239 | 70 | -90 | 2.13 | 36.2 | 1.46 | 218.2 | 0.5-0.9 | 0.12-1.22 |
| 21 | ATD | 194 | 7.6 | 300 | 131 | -120 | 1.7 | 53.9 | 1.60 | 302.3 | 0.6-0.8 | 0.22-1.83 |
| 22 | ATD | 194 | 7.4 | 300 | 131 | -120 | 1.67 | 51.5 | 1.62 | 196.6 | 0.6-0.9 | 0.05-8.44 |
| 23 | ATD | 194 | 7.0 | 250 | 81 | -180 | | | | | | |
| 24 | ATD | 204 | 5.4 | 304 | 74 | -130 | | | | 218.9 | 0.8-13 | 0.07-0.35 |
| 25 | ATD | 204 | 4.9 | 233 | 63 | -100 | 9.98 | 27.8 | 1.2 | 193.4 | 0.4-1.2 | 0.16-1.28 |
| 26 | ATD | 204 | 8.0 | 300 | 131 | -130 | 7.72 | 49 | 1.27 | 351.6 | 0.5-1.1 | 0.06-2.17 |
| 27 | ATD | 204 | 8.1 | 300 | 131 | -160 | 7.58 | 48.4 | 1.07 | 372.9 | 0.8-1.1 | 0.25-0.34 |
| 29 | SA | 194 | 6.5 | 235 | 66 | -140 | 2.04 | 13.2 | 1.84 | 14.9 | 1.4-3.5 | 0.04-7.61 |
| 30 | SA | 194 | 7.6 | 300 | 131 | -140 | 1.65 | 53.4 | 1.88 | 64.5 | 0.6-2.0 | 0.12-0.40 |
| 31 | SA | 194 | 7.5 | 300 | 131 | -150 | 1.78 | 53.3 | 1.95 | 40.3 | 0.8-1.5 | 0.14-0.56 |
| 32 | ATD-SA | 194 | 7.6 | 300 | 131 | -140 | 1.74 | 58.7 | 1.34 | 121.2 | 0.9-1.1 | 0.07-7.49 |
| 33 | ATD-SA | 194 | 7.6 | 300 | 139 | -120 | 1.55 | 60.4 | 1.34 | 259.2 | 0.8-1.5 | 0.15-0.40 |
| 35 | SOA | 189 | 7.3 | 305 | 137 | -140 | 0.73 | 60.9 | 1.88 | 46.8 | 1.0-1.8 | 0.07-5.28 |
| 36 | SOA-HNO$_3$ | 189 | 7.3 | 302 | 134 | -150 | 0.58 | 0.26 | 1.95 | | | |
| 37 | SOA-HNO$_3$ | 189 | 7.2 | 301 | 132 | -140 | 0.79 | 0.47 | 1.34 | 258.9 | 0.8-1.0 | 0.19-22.47 |
| 38 | SOA-HNO$_3$ | 189 | 7.0 | 301 | 135 | -120 | 0.72 | 0.32 | 1.34 | 60.6 | 0.9-1.2 | 0.08-16.96 |
| 40 | ATD | 224 | 7.6 | 234 | 65 | -120 | 129 | 14.8 | 1.18 | 9.3 | 3.1-5.0 | 0.04-3.31 |
| 41 | ATD | 224 | 8.9 | 300 | 134 | -240 | 103 | 31.4 | 1.24 | 17.2 | 4.0-6.1 | 0.07-0.09 |
| 42 | ATD | 224 | 8.4 | 300 | 130 | -160 | 121 | 44 | 1.23 | 15.8 | 3.6-6.5 | 0.06-0.08 |
| 43 | ATD | 224 | 8.5 | 300 | 130 | -130 | 128 | 40.3 | 1.12 | 17.2 | 3.2-6.3 | 0.06-0.08 |
| 45 | SA | 205 | 8.3 | 300 | 134 | -140 | 7.79 | 34 | 1.45 | 18.3 | 1.6-2.9 | 0.12-0.17 |
| 46 | ATD-SA | 204 | 5.5 | 301 | 74 | -130 | 8.32 | 34 | 1.2 | 179.8 | 0.8-1.3 | 0.04-9.89 |
| 47 | ATD-SA | 204 | 5.2 | 233 | 64 | -120 | 10.1 | 27.2 | 1.17 | 177.2 | 0.4-1.2 | 0.03-18.16 |
| 48 | ATD-SA | 204 | 7.6 | 301 | 132 | -150 | 8.06 | 48.5 | 1.12 | 292.8 | 0.9-1.2 | 0.15-1.07 |

limited growth impacts cold mixed phase clouds, and surface diffusion processes may be the limiting growth process in pure ice clouds, as we consider here (Nelson and Baker, 1996; Gierens et al., 2003).

The Knudsen number (Kn) has often been used to evaluate whether ice growth occurs in the kinetic limit (where growth rates are expected to be surface diffusion limited, Kn » 1) or the continuum limit (where they can generally be neglected, Kn « 1) (Pruppacher et al., 1998). Kn is a dimensionless number given by

$$\text{Kn} = \frac{\lambda_w}{R} \tag{5}$$

where $\lambda_w$ is the mean free path of water molecules in air, and $R$ is the particle radius. However, Kn does not provide enough information to assess whether growth rates of facetted ice are surface diffusion limited. Because the deposition coefficient (Eq. 3) varies as a function of $s_{local}$, ice growth may not be significantly limited by surface effects even in the continuum limit, if $s_{local}$ » $s_{crit}$. The relative kinetic to diffusion-limited growth rate as a function of Kn depends strongly on the surface mechanism (e.g., as shown in Figure 14 of Harrison et al. (2016)). In the case of dislocation growth, Kn still generally predicts the ice growth regimes, but for 2D ledge nucleation, ice growth can be significantly suppressed even in the case where Kn « 1.

In Table 2, we show that the IsoCloud experiments have average Knudsen numbers ranging from $\sim$0.01 to 10, suggesting that the majority of ice growth in AIDA occurs in a transitional regime where Kn $\sim$ 1. The Knudsen number is estimated from the average radius (R) of the ice crystals during the experiments. The average mass of each ice crystal is estimated from the total ice water content and the number concentration of ice counted by the OPCs. We determine R by assuming ice crystals are spherical, with a density in $g/cm^3$ given by $\rho_{ice}(T) = p_0 - p_1 T - p_2 T^2$, with $p_0 = 0.9167$, $p_1 = 1.75 \times 10^4$, and $p_2 = 5 \times 10^{-7}$, where T is in Celsius (Pruppacher et al., 1998). The uncertainty in average ice crystal size is estimated to be $\pm$30% based on the combined uncertainties in total ice water content ($2 \times 5\%$) and ice crystal number concentration (20%) (See Table 1). Since ice nucleates over $\sim$30 seconds, ice crystals growing in AIDA have a distribution of different sizes, so this average radius is only an approximation, and there are likely ice crystals that are both smaller and larger than this average in each experiment. In addition to a distribution of different masses, ice crystals in AIDA have previously been observed to have a distribution of particle habits during a single experiment, with a mixture of solid and hollow columns and rosettes observed in expansion experiments at 218 K (Schnaiter et al., 2016).

To quantitatively explore surface kinetics effects during ice growth in AIDA, we use the Lagrangian parcel model with a bin microphysical scheme previously described in Zhang and Harrington (2015). This parcel model uses the DiSKICE model to determine the rate of vapor diffusion to the growing ice crystals, under different assumptions for the deposition coefficient function, as described in Section 2 .

## 4.1 Adaptation of parcel model to AIDA chamber

Lagrangian parcel models typically assume the mass of water is conserved between vapor, ice, and liquid phases and heat is conserved during adiabatic cooling. The AIDA chamber is not in fact a closed system, however, as there is a heat flux through the aluminum chamber walls (2 cm thick, maintained at a constant temperature) and a reservoir of water from ice condensed on the chamber walls. AIDA expansion experiments can be considered pseudo-adiabatic, however. Ice growth during expansion

experiments in AIDA occurs in three distinct phases: an initial nucleation phase, a deposition phase, and a sublimation phase. The nucleation phase sets the condition of supersaturation during the deposition phase, dependent on the availability and type of ice nuclei. Ice growth via deposition occurs while cooling is ongoing, and once the pumps have been turned off, the ice sublimates as the heat flux through the chamber walls warms the gas inside.

To adapt the parcel model to the AIDA cloud chamber experiments, a vapor tendency term was added to the model to include the source of water vapor from the chamber walls. This vapor tendency term is estimated by assuming the change in total measured water (gas plus condensate phases) is due to the flux of water from the chamber walls. Ice crystals > 7 $\mu$m are not sampled with 100% efficiency, so this may be an underestimate however (Cotton et al., 2007). The inclusion of this term significantly improves matching between the model and observations, particularly in the latter part of each experiment when vapor flux from the walls can be significant.

For each experiment, the parcel model is initiated with the observed temperature, pressure, and ice number concentration (Figure 5). Since the OPC's have a 5 s resolution (See Table 1), we use the leading eigenmodes from a singular spectrum analysis (Vautard and Ghil, 1989) on the raw time series to reconstruct the time evolution of the ice number density, as was previously done in Lamb et al. (2017). The initial RH$_i$ measured by the TDL instruments are used to initiate the model. In the parcel model, the bin microphysical scheme divides the ice spectrum into $n_i$ bins, with the number of bins varying with the nucleation rate (up to a maximum of 1000 bins). For the AIDA experiments, we assume that nucleation happens in a linear fashion while the concentration of ice in a volume of air is increasing (as observed by ice particle number concentrations from the OPC's). After nucleation, the ice spectrum evolves by solving the vapor diffusion equations for the growth of the $n_i$ ice bins at each time step using the variable ordinary differential equation package (VODE; Brown et al. 1989). Changes in temperature and pressure are determined directly from the experimentally observed temperature and pressure changes in AIDA, while saturation with respect to ice is calculated in the model. The model assumes the temperature and pressure dependence for the vapor diffusion coefficient $D_v$ according to the Chapman-Enskog theory as given in Seinfeld and Pandis (2016), and the temperature dependence of saturation with respect to ice assumes the parameterization in Murphy and Koop (2005). The parcel model simulates ice and water vapor, as all experiments were performed at temperatures below the homogeneous freezing limit of water (and thus we assume that liquid water would not be present in significant quantities).

Since the ice number concentration during the cooling phase after nucleation is observed to decrease, we infer that some ice crystals are lost to sedimentation or to the walls of the chamber. However, it is not straightforward to estimate the fall speeds of these ice crystals in order to calculate sedimentation rates for each of the $n_i$ bins. We instead account for this ice particle loss by rescaling the number of ice crystals in each of the $n_i$ bins so that the total number of ice crystals summed over all bins is equal to that observed by the OPC's at that time step. The shape of the ice distribution function remains unchanged after this rescaling. Since this assumption would remove ice crystals across size bins at the same rate, rather than preferentially removing larger ice crystals (as might be expected for sedimentation or wall losses), this represents a conservative estimate that may lead to the modeled IWC being biased slightly high. A high bias in IWC would lead to an underprediction of the deposition coefficient when matching the model against the observations.

The model simulates the cooling phase of the experiments while the pressure is decreasing, so as to model only deposition rather than sublimation. In most cases, the sublimation phase can be fit consistently as well, suggesting that similar physics are at play as molecules desorb from the ice. Depending on the length of time that the pumps were on in each experiment, the Lagrangian parcel model simulates between 233-730 seconds. We exclude Exp. 18 (where there was no data for several of the instruments), two experiments with very low IWC (Exp. 19, where IWC was < 0.4 ppmv and Exp. 29 where IWC was < 1.2 ppmv), three experiments at 189 K (Exp. 36-38) where APET had a significant offset from ChiWIS prior to the experiments, suggesting ice may have formed near the extractive inlet, and one experiment (Exp. 39) with very low IN concentrations (< 1 per cm$^3$).

## 5 Model results under different assumptions for surface kinetic effects

To investigate the sensitivity of depositional growth rates in AIDA to the parameterization for the deposition coefficient function $\alpha_D$, we performed several different simulations with the parcel model assuming different models of $\alpha_D$ discussed in Section 2. Since ice crystals are small ($< 20\mu m$, Table 2), here we assume that crystals have not yet developed distinctive habits, i.e. we assume that the deposition coefficients of the a and c axes of the crystals are the same; thus we do not investigate models where $\alpha_D^a$ differs from $\alpha_D^c$. We summarize these simulations below:

- We vary $\alpha_D$ as a constant value (Eq. 2), Section 5.1.

- We vary $\alpha_D$ assuming the updated temperature dependence of $s_{crit}$ from (Harrington et al., 2019), and investigate different surface growth mechanisms (Figure 2), Section 5.2.

- We vary $\alpha_D$ assuming dislocation growth (m=1), under the assumption of various temperature dependent parameterizations for $s_{crit}$ (Figure 3), Section 5.2.

Simulations of experiments performed in the same temperature range and with the same IN demonstrated similar biases with respect to these models. We discuss detailed comparisons of models and observations below.

### 5.1 Sensitivity to surface kinetic effects ($\alpha_D$ = Const.)

We first investigated the average efficiency of ice growth by comparing with different constant values for $\alpha_D$ to see if ensemble ice growth in these conditions can in general be modeled with a constant deposition coefficient, as was assumed in Skrotzki et al. (2013) and has been a typical assumption used in cloud models. By running the parcel model with $\alpha_D$ values of 1.0, 0.2, 0.1, 0.01, and 0.001, we investigated the dependence of the ice growth rates on the deposition coefficient.

Figure 5 shows examples of the parcel model results for 3 experiments performed using ATD to nucleate ice heterogeneously across a range of temperatures (Exp. 4, $T = 230$ K, left column; Exp. 15, $T = 210$ K, middle column; and Exp. 21, $T = 190$ K, right column). The top three panels in each column are the observed temperature, pressure, and ice number concentration, which are used as inputs to the parcel model (the portion of the experiment that is modeled is shown as a red dashed line in

each case). The fourth and fifth panels in each column are observed IWC and $S_i$, respectively, vs. the model results under different assumptions for a constant deposition coefficient. The sixth panel shows the time-dependent value of the deposition coefficient for each simulation (in this case, it is constant while ice is present). In general, the parcel model assuming a constant deposition coefficient agreed best with observations at higher temperatures, and demonstrated the most significant deviations at lower temperatures (< 205 K). Experiments at 235 K (Exp. 1-5) could be consistently modeled with a constant deposition coefficient. Experiments at 224 K (Exp. 7-11) also showed very good agreement with the exception of the initial nucleation phase, where IWC was generally underestimated, and $S_i$ was overestimated by the model. We quantify this further in Section 5.4.

During Exp. 4, there were two separate changes in pressure (from approximately 0-250 s and again from 300 - 750 s in the left panel of Fig. 5), which allowed some of the ice to initially sublimate and then regrow. The modeled IWC and $S_i$ agree well with the higher deposition coefficient models across both the phases of the experiment, indicating that depositional growth and sublimation can be modeled consistently in these conditions. Exp. 4, similar to a significant fraction of the higher temperature experiments (> 220 K) with heterogeneous nucleation on ATD, showed very little sensitivity to $\alpha_D$ for values > 0.2.

Exp. 15 is an illustrative example of an experiment at 210 K with ice growth by heterogeneous nucleation on ATD; these experiments (Exp. 12-17) generally showed good agreement with the model, although the IWC is overestimated by models with $\alpha_D$ for values > 0.2, even though $S_i$ is consistent with the observations, particularly for $\alpha_D = 1$.

Exp. 21 demonstrates the greater deviation of the model from the observations at lower temperatures; similar behavior is observed in the other experiments with ATD at these temperatures (Exp. 20, 22, 32, and 33). Assuming $\alpha_D = 1$ results in lower IWC in some cases than smaller values of $\alpha_D$ (e.g., for Exp. 21, the blue vs green and yellow lines in the fourth panel of the right column in Figure 5), as the very efficient growth depleted the available vapor too quickly, bringing the vapor pressure below saturation and leading to a lower IWC after 200s.

## 5.2 Modeling deposition coefficient with parameterization from Nelson and Baker, 1996

We next investigated ice growth with a deposition coefficient parameterized in terms of a temperature dependent critical supersaturation (Eq. 3), under the assumption of various growth mechanisms. In addition to uncertainty about the surface growth mechanism ($m$ in Eq. 3), the temperature dependent critical supersaturation at temperatures below 233 K is not known. Previous measurements of critical supersaturations for basal and prism facets have been limited to higher temperatures (Nelson and Knight, 1998; Libbrecht, 2003; Harrison et al., 2016; Harrington et al., 2019). Growth rates and habits for ice at temperatures between 203 K and 253 K were experimentally measured in Bailey and Hallett (2004). Even at warmer temperatures, these measurements have significant discrepancies between them (See Figure 3). The functional form for the critical supersaturation ($s_{crit}$) in the DiSKICE model at colder temperatures is not known, as there is not yet consensus about the theoretical basis for this critical supersaturation (Burton et al., 1951; Libbrecht, 2005). The mean displacement of molecules on the surface of a crystal is expected to increase at low temperatures, which means that $s_{crit}$ should generally be inversely correlated with temperature (Burton et al., 1951). Theoretical values are generally derived for the case of ice formation from pure water vapor, although chemical impurities and the existence of quasi-disordered layers could also significantly impact these values

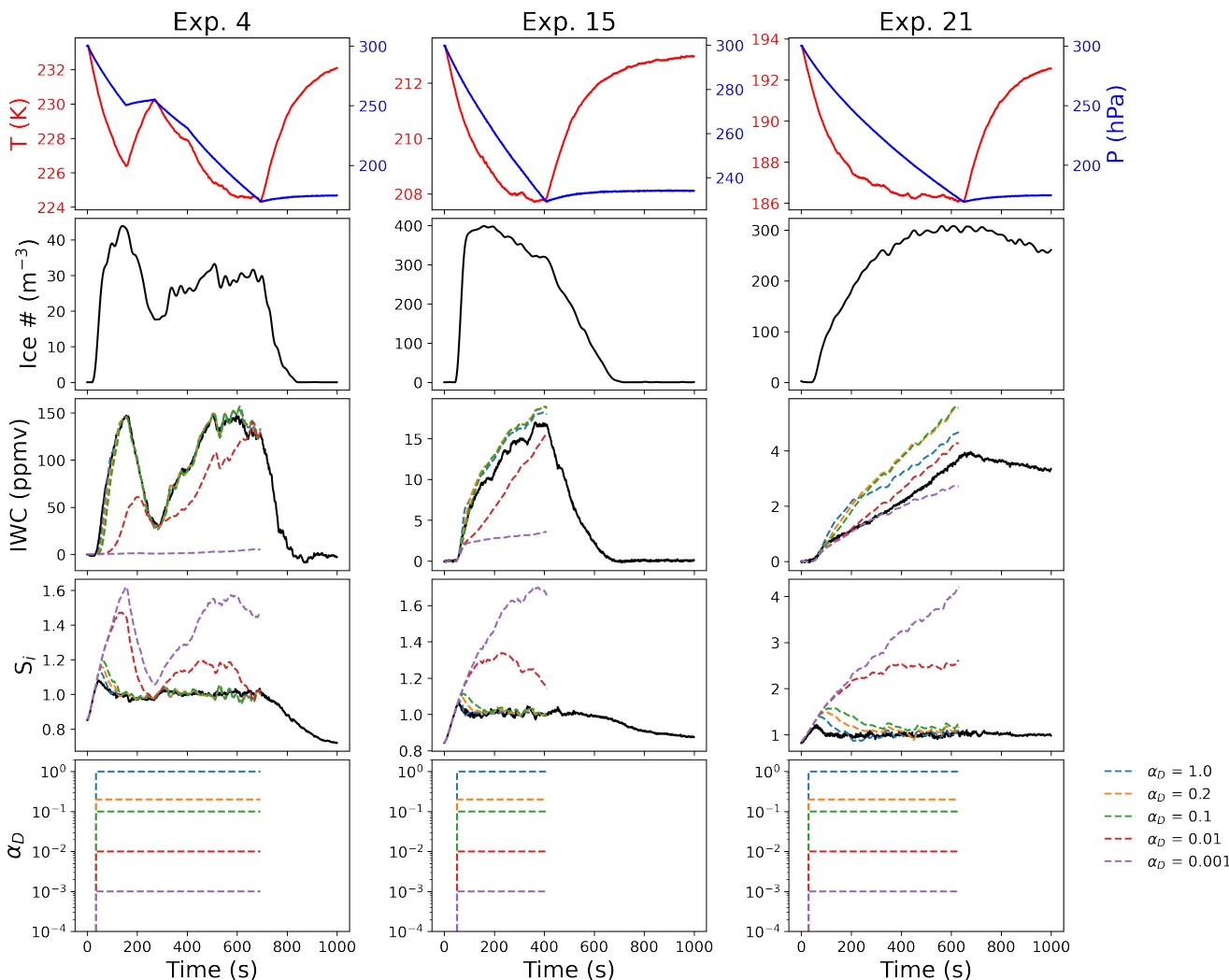

**Figure 5. Parcel model results for different constant values of $\alpha_D$.** Three experiments are shown with different values for constant $\alpha_D$, across a range of temperatures. The modeled IWC in experiment 4 ($T = 230$ K, ATD) is very insensitive to the value for $\alpha_D$ except during the initial nucleation phase. Experiment 15 ($T = 210$ K, ATD) also demonstrates very efficient ice growth, but the modeled IWC shows more significant deviations from the model for $\alpha_D > 0.1$. Experiment 21 ($T = 194$ K, ATD) indicates a case where the IWC is best fit by a constant $\alpha_D$ less than 1; however the observed $S_i$ would not be consistent with the model in this case.

(Libbrecht, 2005). Multiple unknowns still remain in determining an appropriate functional form of $s_{crit}$ for depositional ice growth in the atmosphere, including potential modifications required to account for the impact of nucleation pathway on ice formation.

We investigate the sensitivity of ice growth in AIDA to the surface growth mechanism by varying $m$ (e.g., dislocation growth as $m = 1$, stacking fault dominated growth as $m = 3$, and 2D ledge nucleation as $m = 15$) and by using different

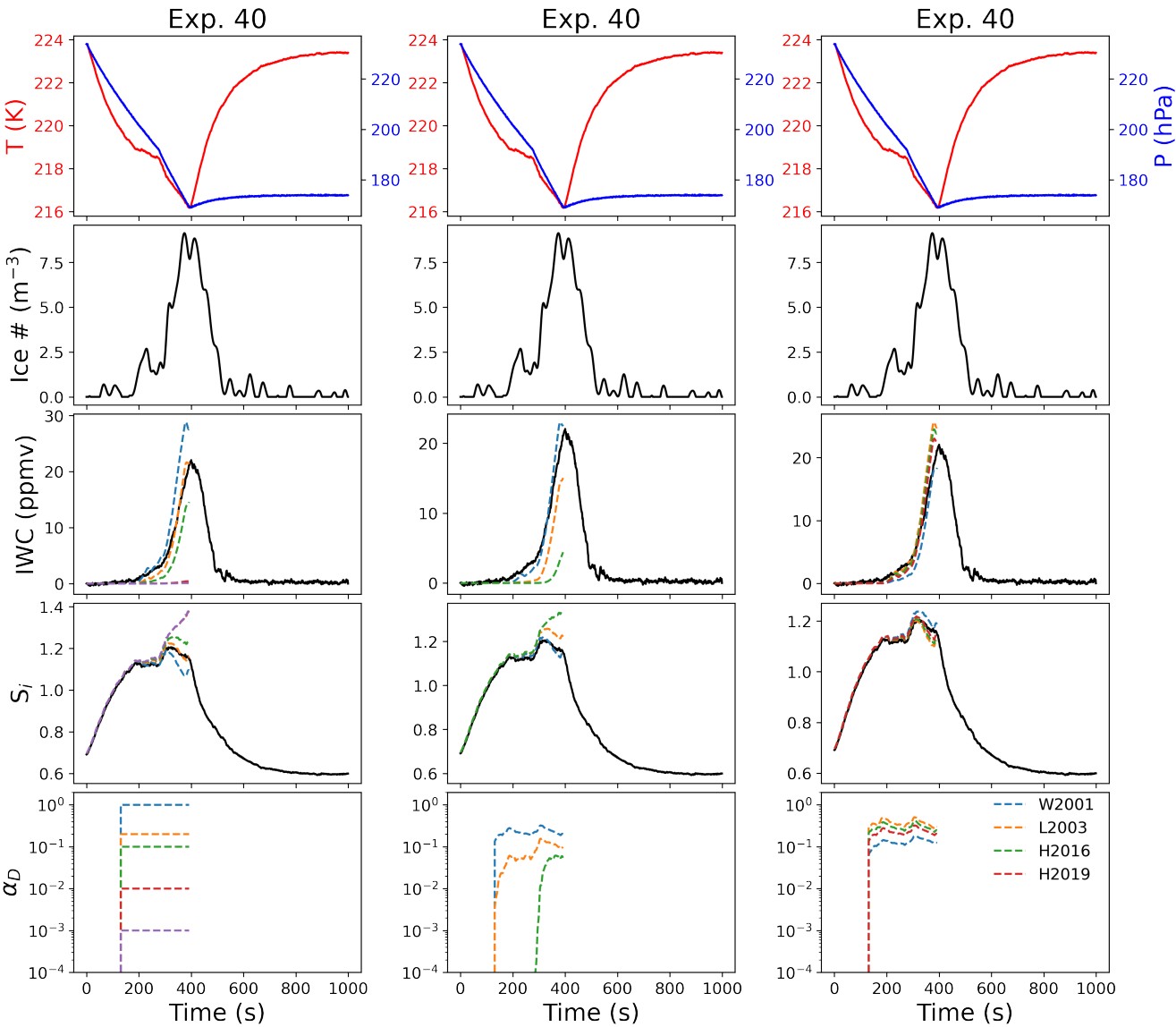

**Figure 6. Comparison between deposition models.** For experiment 40, we show parcel model output for IWC and $S_i$ for different constant values of $\alpha_D$ (left), and for different ice surfaces (middle, varying $m$ in Eq. 3), and for different parameterizations of $s_{crit}$ (right, assuming $m = 1$ in Eq. 3). The top panel in each column shows the temperature and pressure, the second panel the ice number concentration, the third panel the ice water content (as equivalent water vapor mixing ratio), the fourth panel the supersaturation with respect to ice, and the bottom panel the values for $\alpha_D$ in each case. The value for $s_{crit}$ for the model results shown in the middle column is derived from the temperature dependence measured in (Harrington et al., 2019). Here we use the following abbreviations in the legend in the right panel: W2001 (Wood et al., 2001); L2003 (Libbrecht, 2003), H2016 (Harrison et al., 2016), and H2019 (Harrington et al., 2019)

parameterizations for the temperature dependence of $s_{crit}$. Figure 6 shows a comparison between observations and models for Exp. 40 ($T = 220$ K, ATD), for a constant deposition coefficient (left column), different assumptions for surface growth mechanisms (middle column), and different parameterizations for $s_{crit}$ assuming dislocation growth ($m = 1$, right panel). When varying the surface growth mechanisms (middle column), we assume the temperature dependence of $s_{crit}$ derived from Harrington et al. (2019), as experiments were generally most consistent with the lower values of the temperature dependence of the critical supersaturation (as will be discussed in Section 5.4). Exp. 40 had relatively low concentrations of IN present (Table 2).

Exp. 40 was best modeled by a constant deposition coefficient less than 1, with $\alpha_D = 0.2$ (Figure 6 left column). This is similar to the value for $\alpha_D$ predicted for the experiment by Eq. 3, under the assumption of dislocation growth (Figure 6 middle column). The model demonstrated some sensitivity to the temperature dependence of $s_{crit}$ (assuming $m = 1$, Figure 6, right column), with the parameterizations predicting a lower temperature trend more closely matching the observed $S_i$ and IWC. These simulations demonstrate that ice growth during the initial phase of the experiment is more sensitive to the growth mechanism, while later ice growth is more sensitive to the temperature dependence of $s_{crit}$. When we assumed ledge nucleation ($m = 15$) in the model, the model predicted an initially very low deposition coefficient, which caused a significant increase in the simulated $S_i$ in the latter part of the experiment.

## 5.3 Heterogeneous and homogeneous nucleation for experiments at T $\leq$ 205 K

We next compare experiments at 205 K that had different nucleation pathways. Exp. 24-27 were at 204 K and ice formed through heterogeneous nucleation on ATD. Exp. 45 was at 205 K, and was a homogeneous nucleation experiment using SA. Exp. 46-48 had a mixture of both ATD and SA as IN. The homogeneous nucleation experiment (Exp. 45) could be consistently fit with a deposition coefficient near unity or with $\alpha_D$ as given by Eq. 3, assuming $m = 1$ (Figure 7, middle column), whereas the other experiments with heterogeneous IN produced lower IWC than would be predicted by the constant $\alpha_D$ models ($\alpha_D > 0.1$) or by Eq. 3 assuming dislocation growth ($m = 1$). The IWC predicted by Eq. 3 for Exp. 25 and Exp. 47 (assuming stacking-fault dominated growth, $m = 3$) is closer to the observed IWC, but the model predicted significantly higher $S_i$ than was observed in both experiments. Although Exp. 47 included both ATD and SA as IN, this result suggests that the ice growth is mainly controlled by the presence of the heterogeneous IN.

The difference between the heterogeneous and homogeneous nucleation experiments is consistent with the picture that subsequent ice growth in the atmosphere may be strongly influenced by the initial nucleation pathway. During Exp. 45, ice initially grew in highly supersaturated conditions (Figure 7, middle column, fifth panel). Thus, ice growth would not be strongly limited by surface effects in this case, as the high supersaturation required to nucleate ice in the homogeneous nucleation experiment led to very efficient growth.

This effect could be enhanced by differences in ice surface growth mechanisms, as previous observations of surface complexity in AIDA indicated differences between experiments initiated with heterogeneous or homogeneous ice nucleating particles (Schnaiter et al., 2016; Järvinen et al., 2018). In addition, the growth mechanism may change as ice crystals grow or the environmental conditions change. Small ice detector probe (SID-3) measurements of ice formed at 223 K via homogeneous

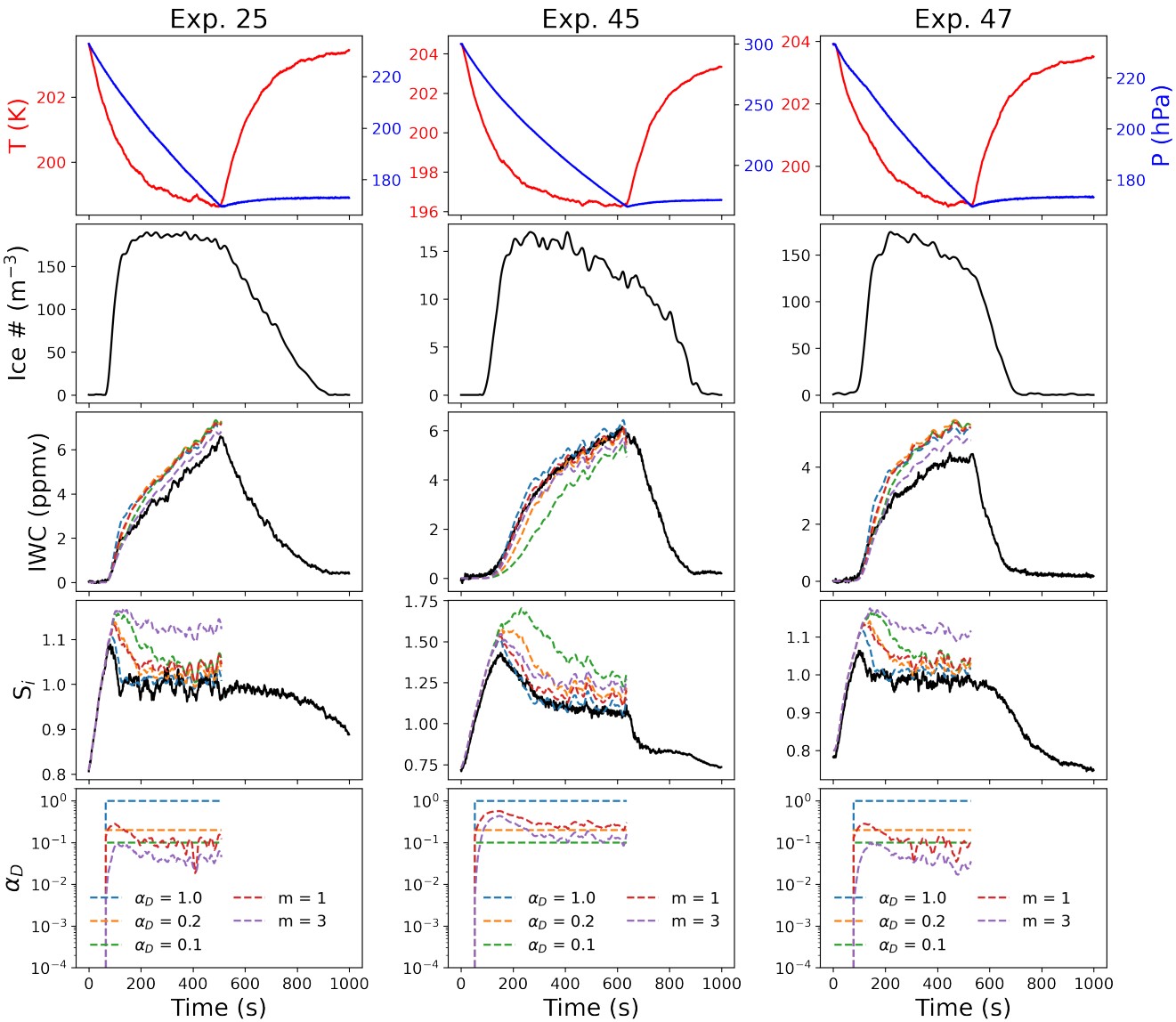

**Figure 7. Comparison between deposition models for experiments at T = 205 K with different IN.** Parcel model output for IWC and $S_i$ for different values of $\alpha_D$ assuming different constant values (1.0, 0.2, or 0.1) or different ice surface growth mechanisms ($m = 1$ or $m = 3$) in Eq. 3. All three experiments took place at similar temperatures, but with different IN: Exp. 25 (ATD), Exp. 45 (SA), and Exp. 47 (ATD+SA).

nucleation in AIDA demonstrated high complexity for these ice crystals; heterogeneous nucleation however was observed to lead to pristine ice (in the case of low supersaturation) and more complex ice for higher supersaturations (Schnaiter et al.,

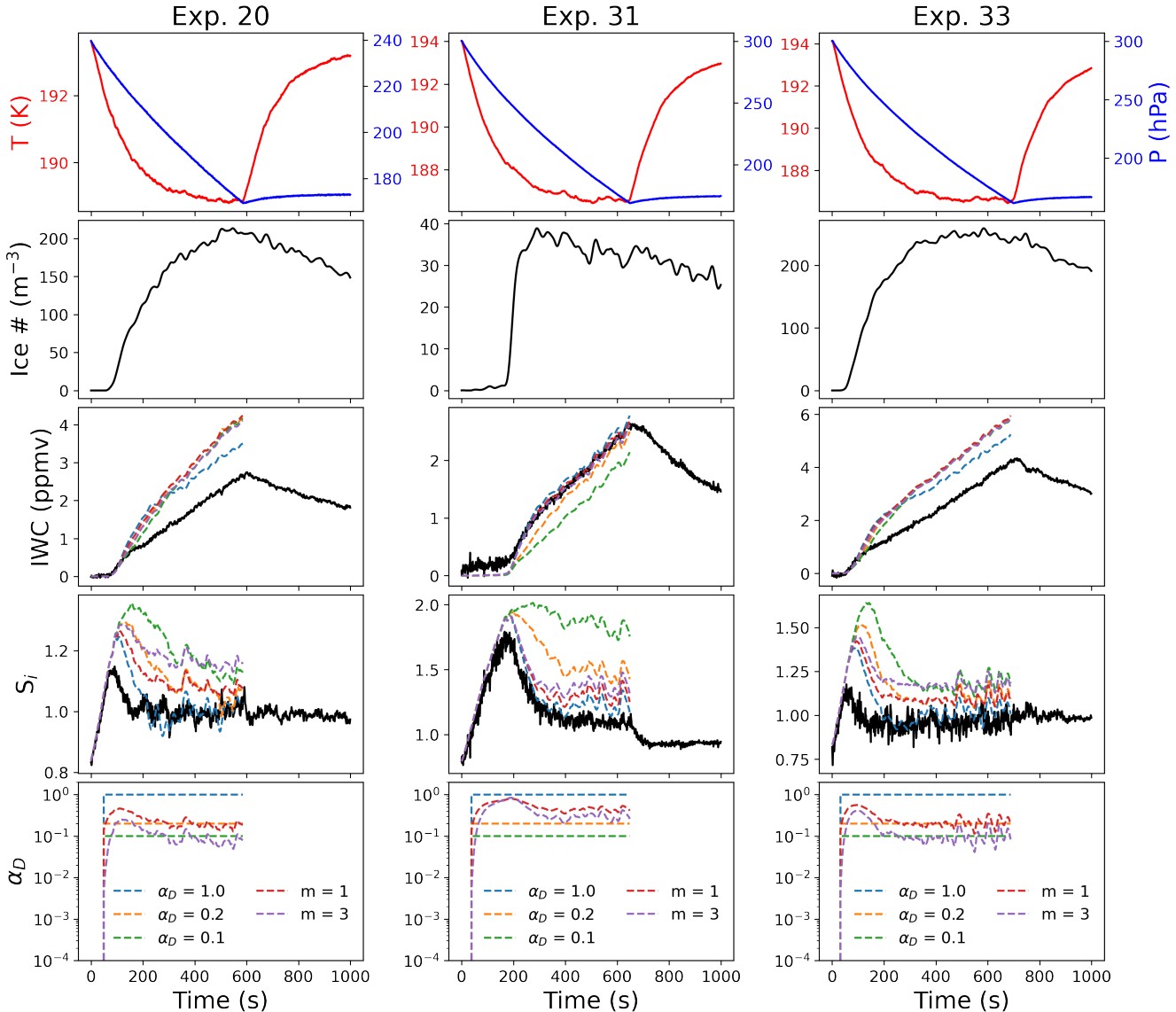

**Figure 8. Comparison between deposition models for cold temperature experiments with different IN.** Parcel model output for IWC and $S_i$ for different values of $\alpha_s$ assuming different ice surface growth mechanisms in Eq. 3. All three experiments took place at similar temperatures, but with different IN: Exp. 20 (ATD), Exp. 31 (SA), and Exp. 33 (ATD+SA).

2016; Järvinen et al., 2018). This suggests that more complex ice in these conditions should be more likely to lead to growth
by abundant surface dislocations.

Comparisons with experiments at lower temperatures ($T = 192$ K) with different IN (Figure 8) demonstrate similar results as those at $T = 205$ K (Figure 7). In this case, Exp. 20 was a heterogeneous nucleation experiment with ATD, Exp. 31 was

a homogeneous nucleation experiment using SA, and Exp. 33 used a mix of ATD and SA. Similar to the experiments near 205 K, the homogeneous nucleation experiment could be fit relatively consistently with the constant deposition coefficient model assuming $\alpha_D = 1$ or Eq. 3 assuming dislocation growth. Both experiments with heterogeneous IN demonstrated a significant mismatch between all models and observations for both $S_i$ and IWC. Varying the temperature dependence of $s_{crit}$ could account for some of the discrepancy between the model and observations, but this result also may be enhanced by an underestimation of the total ice number density for the heterogeneous IN experiments at low temperatures (Clouser et al., 2020). These experiments generally had ice crystals with average radii very near the detection limit of the optical particle counters, and smaller ice crystals on average than were observed for the homogeneous nucleation experiments (Tables 1 and 2). Although models used in the previous study on depositional ice growth in AIDA in Skrotzki et al. (2013) relied on different experimental observables, these models also both used the ice number density as an input, suggesting that the same systematic uncertainty related to undercounting ice at low temperatures would impact those results as well.

## 5.4 Comparison of models and observations as a function of temperature

We use mean absolute percentage error (MAPE) as a metric to evaluate how the model predictions for $S_i$ and IWC for the different simulations compare against the observed time series across all experiments. We adopt notation to define the observed time series for each experiment (either IWC or $S_i$) as $y_i$, and the model prediction as $\hat{y}_i$. MAPE is defined as

$$\text{MAPE} = \frac{1}{n} \sum_{i=1}^{n} \left| \frac{y_i - \hat{y}_i}{y_i} * 100 \right| \tag{6}$$

where the summation is over the $n$ modeled time steps. Because this metric is scale-independent (it is normalized by the number of time steps), it can be used to compare model performance across different experiments. Lower values of MAPE indicate that observations and models more closely agree.

Figure 9 shows the MAPE as a function of temperature for $S_i$ and IWC for the three sets of simulations. The most significant deviations between all models and observations occur below 205 K. For the constant $\alpha_D$ models (Figure 9a,b), the MAPE was generally smallest for $\alpha_D > 0.2$ for $S_i$ across the entire temperature range, and for IWC above 205 K; below 205 K, the MAPE for IWC for some experiments was lower for $\alpha_D < 0.1$.

For the MAPE of $S_i$ and IWC assuming $\alpha_D$ parameterized according to Eq. 3 with different surface growth mechanisms (Figure 9c,d), the experiments above 205 K had the smallest MAPE for both IWC and $S_i$ assuming $m = 1$, or growth by abundant surface dislocations. For experiments below 205 K, the trend was less clear.

Under the assumption of dislocation growth (Figure 9e,f), MAPE for $S_i$ was lower for the assumption that $s_{crit}$ has a lower value at cold temperatures (Figure 3). We plan to further explore statistical constraints placed on the parameterizations for $s_{crit}$ by these experiments in the second part of this study.

To explain why many experiments are consistent with both Eq. 2 and Eq. 3, we note that the radii of the ice crystals changed by a single order of magnitude over the course of these experiments (in most cases from $\sim 1$ to $\sim 5$-$10 \mu m$). This small range of values, coupled with dislocation growth, means that the deposition coefficient function predicted by Eq. 3 is relatively constant

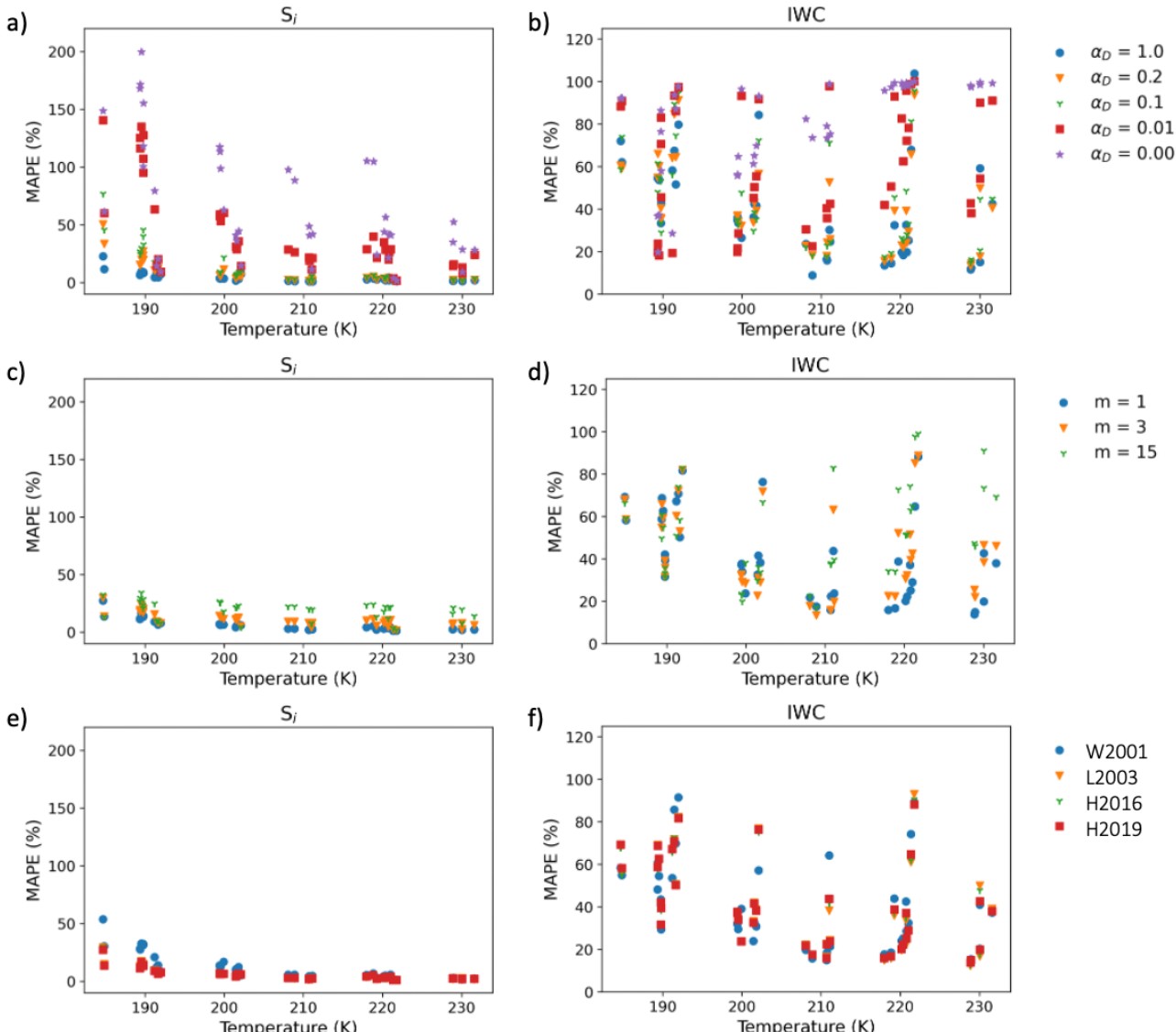

**Figure 9. Mean absolute percentage error for deposition models.** (a,b) Temperature dependence of MAPEs for IWC and $S_i$ for the constant $\alpha_D$ models for all the IsoCloud experiments. (c,d) For the different surface growth mechanisms. (e,f) For different temperature parameterizations for $s_{crit}$ assuming $m = 1$ in Eq. 3. Here we use the following abbreviations in the legend: W2001 (Wood et al., 2001); L2003 (Libbrecht, 2003), H2016 (Harrison et al., 2016), and H2019 (Harrington et al., 2019)

over the experiment. In Exp. 40 for example, the deposition coefficient in the dislocation model ranges from ∼1 to 0.1, which is also a range over which the model is quite insensitive to the deposition coefficient in the constant case (Figure 6).

## 6    Conclusions

Previous experiments in AIDA indicated that the deposition coefficient was within an order of magnitude of unity in cirrus conditions. However, these experiments only considered a model where the deposition coefficient was constant, and these values should be interpreted as an average over the evolving environmental conditions during the expansion experiments. Here we have shown that even though ice growth in cold cloud experiments in AIDA can be modeled with a constant deposition coefficient in most cases, it is also consistent with models of surface kinetic effects that vary as a function of supersaturation and temperature.

Even though these expansion experiments did not generally demonstrate significant depositional growth inhibition due to these surface kinetic effects, we recommend caution in extrapolating from these experiments to real atmospheric conditions. Cloud chamber expansion experiments involve much more rapid changes in supersaturation, temperature, and pressure than are generally found in the atmosphere, and experiments at colder temperatures (< 205 K) demonstrated greater sensitivity to the deposition parameterization, and more significant deviations from all depositional growth models (See Figure 9). Ice growth in AIDA occurs at significantly higher effective updraft speeds (90-370 cm/s for the IsoCloud experiments) than are present in atmospheric cirrus clouds forming in the tropical tropopause layer, where updraft speeds are generally a few cm/s (Krämer et al., 2020). In addition, since populations of ice crystals within AIDA nucleate and grow in relatively consistent conditions, they may not represent the true heterogeneity of ice crystal populations growing in competition with one another in atmospheric conditions. Since surface kinetic effects will be most significant for ice growth from vapor near saturation, the much lower updraft speeds found in regions where *in situ* atmospheric cirrus form indicate these surface kinetic effects should not be neglected in cloud models. Cloud chamber experiments may instead be more representative of atmospheric conditions with significantly higher updraft speeds, such as cirrus clouds formed as the result of overshooting convective systems or in systems with large scale updrafts such as the Asian monsoon (Krämer et al., 2020).

Harrington et al (2019) demonstrated that the low deposition coefficients observed in single particle levitation diffusion chamber experiments (Magee et al., 2006) are consistent with the temperature, super-saturation, and facet-dependent parame-terization of $\alpha_D$ given by Eq. 3. Here we have shown that the high deposition coefficients previously observed in cloud chamber experiments (Skrotzki et al., 2013) can also be explained by this same ice growth theory. Thus, the seeming contradiction be-tween levitation diffusion chamber experiments and cloud chamber studies can be resolved by a non-constant parameterization for $\alpha_D$. However, caution is warranted. While constant deposition coefficients cannot be correct for faceted ice, other processes can affect the growth rates. For instance, immediately following nucleation, facets develop on frozen droplets and grow along the surface. This growth can produce relatively constant mass growth rates (Pokrifka et al., 2020) that can be misinterpreted as a strong kinetic limitation (Harrington and Pokrifka, 2021). This is also true for facet regrowth after sublimation, which can lead to much weaker growth (Harrington and Pokrifka, 2021). Moreover, whether ice forms from heterogeneous nuclei or homogeneous freezing may impact the growth rate (Pokrifka et al., 2020). Since our current theories of ice crystal growth are relatively simple, all surface processes are convolved with the deposition coefficients, and this limitation should be borne in mind when using any deposition coefficient model (Harrington et al., 2019).

Although the AIDA experiments can in some cases be fit with a constant deposition coefficient, the value of the deposition coefficient cannot be predicted directly based on other environmental observables, such as saturation and temperature (as the $s_{crit}$ model in Eq. 3 can be). In a general sense, when $\alpha_D$ is greater than 0.1, attachment kinetics can usually be ignored in mass growth estimates, while $\alpha_D < 0.1$ produces effects on growth that generally cannot be ignored. However, the deposition coefficient in both limits must be included in order to consistently model changes in crystal shape. These results suggest that chamber experiments should not be over-interpreted to imply that the deposition coefficient will always be unimportant for cirrus formation.

The depositional ice growth models described in Section 2 suggest that it is useful to differentiate between the direct influence of the deposition coefficient on mass uptake by the growing ice crystal and the indirect influence of the deposition coefficient on the ice crystal habit. Here, we refer to the direct influence of the deposition coefficient on mass uptake as the "growth efficiency effect" of depositional ice growth. This "growth efficiency effect" is linked to the potential dehydration of water vapor entering the stratosphere through the tropical tropopause layer (Randel and Jensen, 2013). The "habit effect" of depositional ice growth is linked to the direct radiative effects of ice crystals in cirrus clouds. This distinction is also important in developing model parameterizations of depositional ice growth, since the radiative effects of ice crystals and their fall speeds (both of which would be strongly influenced by the "habit effect") are not always consistently treated with the partitioning of water between the vapor and ice phases during depositional ice growth (strongly influenced by the "growth rate effect") in microphysical schemes (Morrison et al., 2020).

These two distinct effects of the deposition coefficient are also linked to typical experimental constraints placed on depositional ice growth. In the cloud chamber experiments described here, it is possible to monitor the partitioning of water between the vapor and ice phases; thus these experiments are directly sensitive to the "growth efficiency effect". In the current study, we did not have information about the ice crystal habits during the growth phase, however. Diffusion chambers use an optical measurement to monitor the growth rates of different facets when crystals are grown on a substrate, and these chambers are sensitive to the "habit effect". In contrast, electrodynamic levitation diffusion chambers monitor the growth rate through changes in the levitation voltage. These devices therefore monitor the partitioning between the vapor and ice phases during the course of the experiment, and sensitivity to the "habit effect" can be discerned through numerical modeling. For instance, levitated crystals formed by heterogeneously frozen pure water drops exhibit growth rates that can only be explained if facets developed on their surfaces (Pokrifka et al., 2020; Harrington and Pokrifka, 2021).

Our study highlights how cloud chamber experiments should be designed to simultaneously monitor the "growth efficiency" effect and the "habit effect" of depositional ice growth. We recommend that future cloud chamber studies focused on depositional ice growth include instrumentation to monitor ice crystal habit during the course of the expansion experiments (for example by monitoring ice growth with the Particle Habit Imaging and Polar Scattering probe or cloud particle imagers (Schön et al., 2011; Abdelmonem et al., 2011, 2016; Schnaiter et al., 2018; Lawson et al., 1998)), as this is necessary to simultaneously constrain both the "habit effect" and the "growth efficiency" in cirrus ice.

Ice growth at low temperatures may be further complicated by a metastable cubic ice phase ($I_c$), which laboratory studies and molecular dynamic simulations have indicated could exist in atmospheric conditions at low temperatures (Murray et al., 2005).

During the IsoCloud campaigns, we saw no evidence for $I_c$ (Clouser et al., 2020). Molecular dynamics simulations at 180 K indicated additional complications for crystal surfaces at low temperatures, including the aggregation of $I_h$ and $I_c$ to form poly-crystals, leading to stacking faults and random grain boundaries (2-dimensional crystal defects) (Moore and Molinero, 2011a, b). Ice crystals with three-fold symmetry (trigonal ice) have also been observed in the atmosphere, and have been related to stacking disorders in the ice crystal lattice (Murray et al., 2015). A recent study exploring experimental water vapor pressure measurements for amorphous solid water and supercooled liquid water crystallizing below 235 K indicates these two distinct phases of water have a phase transition between 200 K and 235 K, which could help explain high supersaturations observed in the TTL (Nachbar et al., 2019). We have not explored these issues here, but they may be important processes in certain atmospheric conditions, and are discussed in the context of the IsoCloud experiments in Clouser et al. (2020).

The IsoCloud experiments provided observational constraints on the values for the critical supersaturation at low temperatures. These experiments indicate that homogeneous nucleation and heterogeneous nucleation in cases of high supersaturation can generally be parameterized as dislocation growth consistent with a temperature dependent critical supersaturation; however ice clouds forming in the atmosphere via heterogeneous nucleation at low ambient supersaturations could proceed via growth mechanisms with a stronger dependence on critical supersaturation (m>1 in Eq. 3). Additionally, changing conditions of supersaturations may contribute to slow growth rates, even when ice was initially nucleated homogeneously (Zhang and Harrington, 2014). More precise laboratory experiments are needed to constrain the distinct growth rates of the basal and prism facets at low temperatures, as the analysis presented here provides only an average growth rate (assuming isometric growth). These experiments suggest that both nucleation pathway (homogeneous or heterogeneous) and ice number density (low or high) may lead to different surface effects. In a follow up paper (Part 2) we will apply Bayesian parameter estimation to more qualitatively evaluate the constraints placed on depositional growth models by the AIDA experiments.

*Code and data availability.* IsoCloud experiment data and python code to preprocess and analyze experiments is available at https://github.com/kdlamb/isocloud. The Lagrangian parcel model code used in this analysis was provided by J.Y. Harrington.

**Table A1. List of acronyms used in the text.**

| | |
|---|---|
| **AIDA** | Aerosol Interactions and Dynamics in the Atmosphere |
| **APeT** | AIDA PCI extractive TDL |
| **APicT** | AIDA PCI in cloud TDL |
| **AquaVIT** | water (aqua) vapor instrumental test |
| **ATD** | Arizona Test Dust |
| **Chi-WIS** | Chicago Water Isotope Spectrometer |
| **DiSKICE** | Diffusion Surface Kinetics Ice Crystal Evolution model |
| **HITRAN** | High Resolution Transmission database for spectroscopy |
| **IN** | ice nuclei |
| **IsoCloud** | Isotopic fractionation in Clouds |
| **IWC** | ice water content |
| **MAPE** | mean absolute percentage error |
| **OPC** | optical particle counter |
| **ppbv** | parts per billion by volume |
| **ppmv** | parts per million by volume |
| **SA** | aqueous sulfuric acid particles |
| **SOA** | secondary organic aerosol |
| **SP-APicT** | single-path APicT |
| **SSA** | singular spectrum analysis |
| **TDL** | tunable diode laser |
| **TDLAS** | tunable diode laser absorption spectroscopy |
| **TTL** | tropical tropopause layer |
| **UTLS** | upper troposphere/lower stratosphere |
| **welas** | Weißlicht-Aerosolspektrometer (White Light Aerosol Spectrometer) |

*Competing interests.* Two of the co-authors (Harald Saathoff and Ottmar Möhler) are members of the editorial board of ACP. The peer-review process was guided by an independent editor, and the authors also have no other competing interests to declare.

*Acknowledgements.* The authors acknowledge support from the U.S. Department of Energy's Atmospheric Science Program Atmospheric System Research, an Office of Science, Office of Biological and Environmental Research program under grant DE-SC0023020. J.Y. Harrington is grateful for support from the National Science Foundation, grant AGS-2128347, and the U.S. Department of Energy's Atmospheric Science Program Atmospheric System Research, an Office of Science, Office of Biological and Environmental Research program under grant DE-SC0021001. We also acknowledge the IsoCloud science team and the AIDA technical staff and support team. Funding for the Iso-

Cloud campaign was provided by the National Science Foundation (NSF) and the Deutsche Forschungsgemeinschaft through International Collaboration in Chemistry grants CHEM1026830 and MO 668/3-1.

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
