# Peer review of "Re-evaluating cloud chamber constraints on depositional ice growth in cirrus clouds—Part 1: Model description and sensitivity tests"

_Atmospheric Chemistry and Physics, 2022_

## Author Comment (AC1)

We thank the two reviewers for taking the time to review the manuscript and for providing detailed comments. We have expanded the text and discussion to address the points the two reviewers raised. In particular, we have made the following changes:

- We have added a discussion in the conclusions on how the results from the current study could inform future experimental designs for constraining depositional growth rates in chamber-based studies.
- We have expanded the explanation on how the bin microphysical model described in Zhang and Harrington, 2015 was adapted to AIDA.
- We have added a section to draw a clear distinction between what we term the depositional ice growth "habit effect" and the "growth efficiency effect".
- We have added a table of the acronyms used in the text to Appendix A1.

These changes have significantly strengthened the manuscript and clarified the scope of our current study. Where possible, we have also made the figures larger and easier to read and reduced some unnecessary columns in tables so that they are larger.

Below we give point by point responses to each of the comments raised by the reviewers, with the reviewers' initial comments in black and our responses in blue, with changes made to the manuscript highlighted in green. Line numbers refer to the revised manuscript. In addition to addressing the reviewers' comments, we have corrected a few typos in the text.

General comments.
The study provides an experimental test of the influence of the deposition coefficient functions on small ice crystals in simulated cirrus conditions. Unlike previous studies at such cold temperatures (below -40 C), the experiments here are run in a large cloud chamber with many crystals growing simultaneously, and the data are compared to an advanced microphysics model of ice growth. For results, they find that supersaturation-dependent functions best fit the data, though the values are relatively large. The findings should be useful to researchers of cirrus and should be published.

The weak points of the approach, which probably should be made clearer in the text, are the apparent lack of verification that the crystals have facets (e.g., by sampling or some optical method) and the short growth times (typically about 10 min or less). Sure, the experimental conditions make these improvements difficult, though they seem like obvious steps to attempt in future studies and worthy of mention. About the facets, I would expect that small, nearly spherical crystals would have deposition coefficient functions near unity (surface roughness, higher local supersaturation) and thus not very appropriate for the theoretical model, whereas facetted crystals would have smaller deposition coefficient values that would depend on the face. So, verifying that facets exist seems important for applying the model. About the need for longer growth times, longer growth at lower supersaturations may simulate actual cirrus better and also provide a better test to the theoretical model.

Other than those issues, the approach described here looks fine, though I have some other general suggestions to help clarify the study.

We thank Dr. Nelson for taking the time to provide very detailed and thorough comments, and agree with his points about additional experimental needs that would be useful in future cloud chamber studies to put stronger constraints on depositional ice growth at UTLS temperatures.

We do have one comment on the weakness of the model. We do agree that the model has a fundamental weakness in that it assumes facets appear essentially instantaneously on the crystal surface. In reality, the crystals are most likely developing facets over time, such as in the work by Gonda and Yamazaki (1984, J. Crystal Growth).  Some of our prior experimental work using the levitation diffusion chamber produced growth rates consistent with crystals that undergo facet development (Pokrifka et al., 2020). Using these experiments along with ideas from Nelson and Swanson (2019, Atmos. Chem. Phys.), we showed in Harrington and Pokrifka (2021, J. Atmos. Sci) that growth rates similar to the experiments could be explained with a simple theory that includes rough and faceted regions. The time required for facet development was a few minutes, so the crystals in AIDA may have been within this growth regime. If so, then a rough growth model would not be appropriate, but a faceted model wouldn't be entirely appropriate either. On the other hand, given the rapid cooling rates and commensurately rapid rise in supersaturation, any transition from rough to faceted ice could occur quickly. If so, then the model of growth used here (isometric crystals grown by dislocations) is arguably appropriate. Nevertheless, given that rough and faceted regions probably occurred on the same

crystals, a model of growth that combines both regions may be needed to simulate the growth of crystals in the AIDA chamber, and this should be explored in future work. It is worth noting that the real situation is probably even more complex. We imagine that, at the high supersaturations developed in AIDA, hollowing and the development of branching arms may occur, which would then complicate the growth picture. In previous studies in AIDA (Schnaiter et al. 2016, ACP), ice particle replicas generated during expansion experiments at -55 C (218 K) indicated clear evidence of solid and hollow columns and budding rosettes. In a sense, the analysis that we have done here assumes that the crystals instantaneously develop facets but also never experience any morphological changes. As a consequence, the deposition coefficients we derive may conflate rough growth and faceted growth. They should therefore be viewed as coefficients that parameterize the overall growth process for newly formed ice.

To address this comment and also the later point about the direct and indirect effects of deposition on growth rates and ice crystal habits, we have expanded the Conclusions Section to include a discussion of how the limitations of the current study could be addressed in future experiments. We have added the following paragraphs to the conclusions in lines 523-542:

"The depositional ice growth models described in Section 2 suggest that it is useful to differentiate between the direct influence of the deposition coefficient on mass uptake by the growing ice crystal and the indirect influence of the deposition coefficient on the ice crystal habit. Here, we refer to the direct influence of the deposition coefficient on mass uptake as the "growth efficiency effect" of depositional ice growth. This "growth efficiency effect" is linked to the potential dehydration of water vapor entering the stratosphere through the tropical tropopause layer (Randel and Jensen, 2013). The "habit effect" of depositional ice growth is linked to the direct radiative effects of ice crystals in cirrus clouds. This distinction is also important in developing model parameterizations of depositional ice growth, since the radiative effects of ice crystals and their fall speeds (both of which would be strongly influenced by the "habit effect") are not always consistently treated with the partitioning of water between the vapor and ice phases during depositional ice growth (strongly influenced by the "growth rate effect") in microphysical schemes (Morrison et al., 2020).

These two distinct effects of the deposition coefficient are also linked to typical experimental constraints placed on depositional ice growth. In the cloud chamber experiments described here, it is possible to monitor the partitioning of water between the vapor and ice phases; thus these experiments are directly sensitive to the "growth efficiency effect". In the current study, we did not have information about the ice crystal habits during the growth phase, however. Diffusion chambers use an optical measurement to monitor the growth rates of different facets when crystals are grown on a substrate, and these chambers are sensitive to the "habit effect". In contrast, electrodynamic levitation diffusion chambers monitor the growth rate through changes in the levitation voltage. These devices therefore monitor the partitioning between the vapor and ice phases during the course of the experiment, and sensitivity to the "habit effect" can be discerned through numerical modeling. For instance, levitated crystals formed by heterogeneously frozen pure water drops exhibit growth rates that can only be explained if facets developed on their surfaces (Pokrifka et al., 2020; Harrington and Pokrifka, 2021)."

1.  As the experiments were done in 2013, I wondered if the data had been previously analyzed and published. If so, where and how does the present analysis differ?

The observations from these experiments were previously analyzed in Lamb et al. 2017 and Clouser et al. 2020 (as described in Section 3 on p. 8, lines 192-194), although these previous studies did not focus on evaluating depositional ice growth models, but rather investigated isotopic fractionation between HDO and H2O (in Lamb et al. 2017) and compared saturation vapor pressure with respect to ice during the sublimation phase of the expansion experiments with the Murphy and Koop parameterization for saturation with respect to ice (in Clouser et al. 2020). As discussed in the text, a previous study (Skrotzki et al. 2013) did investigate depositional ice growth in AIDA, using a different set of experiments. In addition, the current version of the DISKICE model was developed starting with Zhang and Harrington 2014, 2015 and further developed in Harrington et al. 2019. These more recent modeling studies have provided context for reanalyzing the past experimental data sets in light of the updated theoretical understanding– the focus of this current study.

We added this background to Section 3, lines 201-203:
"Previous studies using the IsoCloud observations focused on characterizing isotopic fractionation between HDO and H$_{2}$O during depositional ice growth (Lamb et al. 2017) and investigated saturation vapor pressure over ice for temperatures between 189 and 235 K (Clouser et al. 2020)."

2.  The method of determining crystal sizes is not clear. I found the method described in the Skrotzki et al. 2013 paper, though it would help to clarify it here. Also, it seems like a rather indirect method to extract crystal size, so it may help to discuss the uncertainties.

We have more clearly spelled out the method used to determine the crystal sizes in lines 286-290.

"The average mass of each ice crystal is estimated from the total ice water content and the number concentration of ice from the OPCs. We determine R by assuming ice crystals are spherical, with a density in $g/cm^{3}$ given by $\rho_{ice}(T)=p_{0}-p_{1}T-p_{2}T^{2}$, with $p_{0}=0.9167$, $p_{1}=1.75e04$, and$ p_{2}=5e-7$, where T is in Celsius (Pruppacher and Klett, 1997). The uncertainty in average ice crystal size is estimated to be ±30% based on the combined uncertainties in total ice water content (2x5%) and ice crystal number concentration (20%) (See Table 1)."

3.  How much spatial variability exists in the supersaturation and crystal sizes? Related to this issue, if crystals have a range of sizes (and shapes), how would this affect the application of the theoretical model? A related question is variability in the deposition coefficient functions even when all crystals are the same size, shape, and have the same local supersaturation—how would that affect the application of the theory? Of course, these cannot be answered exactly, though it seems worthwhile to estimate the potential influence on the results.

We clarified this point in lines 290-295:

"Since ice nucleates over ~30 seconds, ice crystals growing in AIDA have a distribution of different sizes, so this average radius is only an approximation, and there are likely ice crystals that are both smaller and larger than this average in each experiment. In addition to a distribution of different masses, ice crystals in AIDA have previously been observed to have a distribution of particle habits during a single experiment, with a mixture of solid and hollow columns and rosettes observed in expansion experiments at 218 K (Schnaiter et al., 2016)."

In the bin model, we assume that the ice has a distribution of different sizes, as is described in more detail in the response to the comment related to Line 271 (see below). After nucleation, each crystal bin evolves independently. We clarify this point in the text in lines 315-339.

4. Are there any sharp images of crystals of these small sizes grown under these conditions (either from a cloud chamber or an actual cloud)? It would help to show them, or an accurate sketch, in the Introduction.

Some examples of ice grown in AIDA at -55C (218 K) can be seen in Figure 5 of Schnaiter et al. 2016, and show small "budding" rosettes as well as solid and hollow columns with sizes ~20 - 35 um. We added this point to the text in lines 293-295.

"In addition, ice crystals in AIDA have previously been observed to have a distribution of different particle habits during a single experiment, with a mixture of solid and hollow columns and rosettes observed in expansion experiments at 218 K (Schnaiter et al. 2016)."

5. The growth model is also dependent on the vapor diffusion coefficient. What values were used? As the modeled rate can be more sensitive to this parameter than the deposition coefficient functions, perhaps you could make use of the sublimation phase (when it would be safe to assume a sublimation coefficient of unity) to measure the appropriate vapor diffusion constant. The values so derived would then be input into the growth model.

We calculate the vapor diffusion coefficient using the traditional model assuming Chapman-Enskog theory as given in Seinfeld and Pandis, 2016, but there is uncertainty with regards to this value at these temperatures. We have added this point to the text in lines 325-327. We also clarify that we are assuming the Murphy and Koop parameterization for the temperature dependence of saturation with respect to ice. Both of these parameterizations have some uncertainty, particularly at these low temperatures– this is a point we plan to explore in Part II of this work.

6. In looking at the results in Figs. 5-8, what is thought to be the main factor causing the oscillations and bumps in the ice-crystal number? Influence from the pumping, sticking to walls, sampling for IWC, new crystal nucleation, or just noise in the measurements?

Two factors play a role in the oscillations and bumps in the ice crystal number. The first is the inhomogeneity across the chamber in terms of temperature and saturation fluctuations (as can be seen in the observed temperature and saturation trends during expansion experiments in

Figures 5-8). The saturation with respect to ice is derived from the measurement of water vapor via direct absorption across the entire chamber volume (about 2/3rds of the height of the chamber, see Figure 4), and thus the small fluctuations in saturation represent real variability in the cloud across the chamber. Inhomogeneities in the chamber gas are also observable in analysis of the TDLAS alignment beam stability, which points to small index of refraction changes in the gas (due to Rayleigh scattering, see Sarkozy et al. 2020, Section IIIC and Figure 7.)

The second factor is the statistical noise in the ice crystal number concentration from the OPC's– particularly for cases where the absolute ice number concentration is relatively low, fluctuations on the order of a few particles per m-3 are not unexpected.

7. It might help readers if there is a table defining the many acronyms.

We added a table to Appendix A defining the acronyms used in the text.

8. The paragraph speculating about cubic ice in the conclusions could be removed. I have yet to see any convincing evidence for cubic ice in the atmosphere, though stacking faults in vapor-grown ice Ih seem to be well established.

In AIDA we saw no evidence for Ic, although since this has been discussed numerous times in the literature, we feel that it is appropriate to include this point in the conclusions. We have added this clarifying point to the text in lines 551:

"During the IsoCloud campaigns, we saw no evidence for $I_{c}$ (Clouser et al. 2020)."

Smaller comments and suggestions, by line number.

Line 3. May be better to always distinguish the "direct" influence of deposition coef. functions on mass uptake vs the "indirect" influence via crystal shape. In some places in the text, this is made clear, though would help here to add "direct" or some similar clarifying term.
This is a great point, since this more clearly differentiates the impact of these two effects relevant for the impact of depositional ice growth in cirrus clouds on climate. In the abstract, we clarified line 3 to read:

"Surface attachment kinetics, generally parameterized as a deposition coefficient $\alpha_{D}$, control ice crystal habit and also may limit growth rates in certain cases, but significant discrepancies between experimental measurements have not been satisfactorily explained."

We have also added a discussion in the conclusions, to clearly define the "growth efficiency effect" as compared to the "habit effect" of depositional ice growth. We have chosen to adopt these two terms, rather than "direct" and "indirect" to avoid any confusion with "direct" and "indirect" climate effect (as the "growth rate effect" is more of an "indirect" effect in terms of its climate impact and the "habit effect" is more closely linked with the "direct radiative effects" of cirrus clouds).

Line 6. Can you break up the sentence to ease reading?

Fixed.

Line 10. Are all models declared proper nouns? I know capitalized models commonly appear in the literature, though that doesn't automatically make it correct usage. Maybe I'm wrong here, though to some readers, quotation marks look better for the long model names.

We have adapted the name DiSKICE as it was previously introduced in Harrington et al., (2019).

Line 14. Maybe add that the crystal is assumed spherical, or at least, isometric?
We have updated this sentence to read "DiSKICE model assuming growth on isometric crystals via abundant surface dislocations"

Line 27. It looks like the authors of the 2018 paper use the term "ice crystal complexity" instead of "surface complexity". It also seems more appropriate.
Fixed.

Line 37. Perhaps "constant" should be specified further to mean independent of supersaturation, temperature, and facet?
Added.

Line 42. And dependent on crystal facet.
Updated.

Line 42. Here and elsewhere, the Latin abbreviations commonly are followed by a comma "e.g.,"
Updated.

Line 70. It might help here to specify whether you are referring to the limiting (high-supersaturation value) of the deposition coefficient or the functional form of the deposition coefficient function. Concerning the former, I could explain quite a few cases of the limiting (high-supersaturation) deposition-coefficient function discrepancies of not just ice, but also rare-gas crystals, as arising by heat conduction effects (J. Cryst. Gr. vol. 132 (1993) 538—550.) At the time (pre-1993), many studies of other crystals had made it clear that the limiting case, also known as the "sticking coefficient", was always equal to unity for crystal growth (unless a chemical reaction was involved). Yet there seemed to be many values claimed for ice that were much less.
This is a good point, and some of the confusion in the experimental literature around this point appears to be that the high-supersaturation limit for the deposition coefficient was used in cases where the ice growth was near saturation with respect to ice. We clarified that the previous AIDA study focused on whether depositional ice growth rates were limited by surface effects (effectively fitting a constant value for the deposition coefficient) in lines 69-75:

"The past study in AIDA focused on whether surface kinetic effects limit ice growth rates in cirrus conditions, effectively investigating models for the deposition coefficient that assume it is a single constant value, rather than a supersaturation, temperature, and facet dependent function. This constant value for the deposition coefficient can be thought of as the high-supersaturation limit of the deposition coefficient function. Calculating the shape evolution of faceted crystals, even complex ones, would still require estimating parameters for the temperature, supersaturation, and facet-dependent deposition coefficient function, even if this deposition coefficient function does not significantly limit growth rates."

Line 73. Here might be a place to mention prior analyses of the experiments (see the general comments). When I read the first two sentences, I thought you were re-examining the Skrotzki et al. 2013 experiments, though those must have been before 2013 and also involved only 15 experiments, much less than your 48.

We have reworded these sentences to make it clearer, that while Skrotzki et al. 2013 also investigated depositional ice growth in the context of AIDA experiments, here we are focusing on a different set of experiments that were previously discussed in Lamb et al. 2017 and Clouser et al. 2020.

"To address these questions, we use observations from expansion experiments performed inside of the AIDA Aerosol and Cloud chamber during the IsoCloud (Isotopic Fractionation in Clouds) campaign (Lamb et al. 2017, Clouser et al. 2020). We investigate whether depositional ice growth models including surface kinetic processes that vary with changes in ambient conditions are consistent with the observed ice growth rates in AIDA. These experiments included cases of both homogeneous and heterogeneous ice formation to investigate the role of ice nucleation pathways on depositional growth. While Skrotzki et al (2013) explored the variability in the deposition coefficient derived from experiments performed in AIDA …"

Line 96. Here, and I think a few other places, you refer to a textbook or review article instead of the original authors. It seems fairer to credit the original authors of the idea. In this case, it may be Fukuta and Walter, 1970, or even earlier. Definitely not Pruppacher et al. 1998.
Updated.

Line 100. It looks like Si is the vapor supersaturation ratio.
Updated.

Line 121. "considered to be unity for nearly all crystals." See 70. Above.
We clarified this point as discussed in the comment above on line 70.

Line 122. There were earlier molecular-beam experiments that found unity. (Sorry, I don't recall the references, though maybe by D. Kay et al. in the early 90s or earlier.) Also, the claim that the sticking coefficient decreases with increasing temperature is a signature of heat conduction effects analyzed in 70. Above.

We added this point to the text in lines 129-130:
"Experiments using a molecular beam indicated that this sticking coefficient $\alpha_{s} \sim 1$ and is temperature-independent (Brown et al. 1996)."

Line 135. Gonda et al. did advanced optical microscopy much earlier, giving strong evidence for growth spirals.
The papers we referred to here demonstrated direct observation of these growth mechanisms and contained references to the previous work (such as Gonda et al.) that provided more indirect evidence. We have clarified this sentence in line 141 to read:

"See for example, Sazaki et al. (2010, 2014) and references therein."

Line 154. Very minor quibble about terms here. It is unclear whether the term "instability" is useful for the branching case on snow crystals. Perhaps the best description of branch formation on snow crystals is from F.C. Frank (Contemp. Phys., 1982), and he had no need for the term. "Instability" applies to the melt-grown ice dendrites, arising from the Mullins-Sekerka study, though vapor-growth is very different from the melt-growth case. Hollowing at low supersaturations seems unstable, judging by the various asymmetries between facets, though this hasn't been studied in any detail. It just seems unnecessary to use the term here.

The reviewer makes a great point that Frank never needed to use "instability" to describe branching, but other authors have employed the term when describing the tendency of crystals to develop both hollowed regions and branches at higher supersaturation. For instance, Yokoyama and Kuroda (1990, Phys. Rev. A) used the phrase "morphological instability" to describe the process of branching growth above a transitional supersaturation. Those authors were careful to distinguish this branching instability from then Mullins-Sekerka instability that occurs when crystals grow from the melt. We like the phrase "morphological instability", because it is compact and indicates that a transition in crystal habit has occurred. We have clarified that branching and instability on facets may also be important in lines 158-162:

"For $l_{h}$, the basal and prism facets have been shown to have different deposition coefficients, with distinct dependencies on temperature and supersaturation, consistent with the large variety of ice crystal morphologies observed in the atmosphere. Once deposition coefficients reach unity, morphological instabilities can occur leading to the hollowing of facets and the development of branching arms (e.g. Gonda and Gomi, 1985; Yokoyama and Kuroda, 1990; Wood et al., 2001; Libbrecht, 2005). "

Line 170. This sentence seems awkward. Perhaps move the "prior to nucleation" part to later in the sentence.

Updated to:
"Experiments on heterogeneous ice nuclei have demonstrated surface defects on mineral dust serve as active sites for ice growth (Kiselev et al., 2016).

Line 221. This sentence is rather wordy. I think you mean "A mixing fan inside the chamber pushes the crystals up and mixes the air." However, it seems just as much air is pushed down as up. Also possibly relevant: for the estimated crystal sizes here, it seems that settling would let crystals fall by only about a meter (~1/5$^{th}$ the chamber height) over the course of the roughly 10-minute experiments. So maybe the "mixes" is useful here, not the "pushing up"?

Table 2. It would help to define the variables in a table footnote, also give the units for CCN, the meaning of ATD, SA, SOA, as well as the meaning of the range in R. Some of these can be found in the text, but not all in the same spot.

The mixing fan is located at the bottom of the chamber (as can be seen in Figure 4). Thus the main action of the fan is to push air up and to reduce the action of gravitational settling. We have clarified this sentence to read:

"A mixing fan located at the bottom of the chamber mediates the impact of gravitational settling and decreases inhomogeneity in the gas."

As an aid to the reader, we have added a table of acronyms in the appendix (Table A1) and added a reference to this table in the text in lines 98:

"Since a number of acronyms are used throughout the text, we provide a reference list in Table A1."

Line 255. You use growth and evaporation here, but elsewhere deposition and sublimation. Updated to be consistent.

Line 265. Perhaps clarify that the growth rate could be surface limited at any Knudsen number, though certainly when it is much greater than unity.

We have added the definition of the Knudsen number, and clarified the discussion in lines 274-286:

"The Knudsen number (Kn) has often been used to evaluate whether ice growth occurs in the kinetic limit (where growth rates are expected to be surface diffusion limited, Kn » 1) or the continuum limit (where they can generally be neglected, Kn «1) (Pruppacher et al., 1998). The Knudsen number (Kn) is a dimensionless number given by
\begin{equation}
Kn = \frac{\lambda_{w}}{R}
\end{equation}
where $\lambda_{w}$ is the mean free path of water molecules in air, and $R$ is the particle radius. However, Kn does not provide enough information to assess whether growth rates of facetted ice are surface diffusion limited. Because the deposition coefficient (Eq. 3) varies as a function of $s_{local}$, ice growth may not be significantly limited by surface effects even in the continuum limit, if $s_{local}$ » $s_{crit}$. The relative kinetic to diffusion-limited growth rate as a function of Kn depends strongly on the surface mechanism (as shown in Figure 14 of Harrison

et al. (2016)). In the case of dislocation growth, Kn still generally predicts the ice growth regimes, but for 2D ledge nucleation, ice growth can be significantly suppressed even in the case where Kn « 1.

In Table 2, we show that the IsoCloud experiments have average Knudsen numbers ranging from ~0.01 to 10, suggesting that the majority of ice growth in AIDA occurs in a transitional regime where Kn ~ 1. The Knudsen number is estimated from the average radius (R) of the ice crystals during the experiments."

Line 271. It might help to define a bin here. The term comes up again in a few paragraphs.

We clarify that the parcel model uses a bin microphysics model to investigate how the spectrum of ice in a volume of air evolves during the course of the expansion experiment. We added the following paragraphs to the text in lines 315-339 to clarify how the bin microphysics scheme works.

"In the parcel model, the bin microphysical scheme divides the ice spectrum into $n_{i}$ bins, with the number of bins varying with the nucleation rate (up to a maximum of 1000 bins). For the AIDA experiments, we assume that nucleation happens in a linear fashion while the concentration of ice in a volume of air is increasing (as observed by ice particle number concentrations from the OPC's). After nucleation, the ice spectrum evolves by solving the vapor diffusion equations for the growth of the $n_{i}$ ice bins at each time step using the variable ordinary differential equation package (VODE; Brown et al. 1989). Changes in temperature and pressure are determined directly from the experimentally observed temperature and pressure changes in AIDA, while saturation with respect to ice is calculated in the model. The model assumes the temperature and pressure dependence for the vapor diffusion coefficient $D_{v}$ according to the Chapman-Enskog theory as given in Seinfeld and Pandis (2016), and the temperature dependence of saturation with respect to ice assumes the parameterization in Murphy and Koop (2005). The parcel model simulates ice and water vapor, as all experiments were performed at temperatures below the homogeneous freezing limit of water (and thus we assume that liquid water would not be present in significant quantities).

Since the ice number concentration during the cooling phase after nucleation is observed to decrease, we infer that some ice crystals are lost to sedimentation or to the walls of the chamber. However, it is not straightforward to estimate the fall speeds of these ice crystals in order to calculate sedimentation rates for each of the ni bins. We instead account for this ice particle loss by rescaling the number of ice crystals in each of the $n_{i}$ bins so that the total number of ice crystals summed over all bins is equal to that observed by the OPC's at that time step. The shape of the ice distribution function remains unchanged after this rescaling. Since this assumption would remove ice crystals across size bins at the same rate, rather than preferentially removing larger ice crystals (as might be expected for sedimentation or wall losses), this represents a conservative estimate that may lead to the modeled IWC being biased slightly high. A high bias in IWC would lead to an underprediction of the deposition coefficient when matching the model against the observations."

Line 291. The "assumption" did not seem clear, and perhaps the consequence could be explained better.

The ice number concentration is observed to decrease (even while the cooling phase of the experiment is still on-going). From this we infer that some ice crystals are lost to sedimentation or to the walls of the chamber. Thus we constrain the number concentration to be the experimentally observed value (as we explained in the preceding response). We tried other methods (such as removing ice at a constant rate) but this approach provided the most consistency between observations and models.

Line 296. Perhaps explain that you are assuming dry adiabatic ascent to deduce the updraft to use in the model? Or, am I not understanding this? Then, on 299, I suppose by "Typical updraft" you mean this "effective vertical motion"?

We have clarified these points in lines 228-233. We moved this paragraph to Section 3, since the dry adiabatic ascent is only used to estimate the effective updraft speeds. The model is directly constrained to observed temperature and pressure and thus does not make direct use of the updraft speed.

Line 312. From the table, it looks like the largest size is less than 14 microns. Also, on this line, I think you mean "we assume that crystals remain isometric". Developing a non-isometric habit takes time, so "have not yet developed" doesn't mean all crystal faces grow at the same rate.

We have clarified here that these sizes are the average effective spherical radius for the ice crystals estimated from the ice mass mixing ratio and number concentration, assuming all ice is the same size– we do not believe that this accurately describes the true ice crystal size in AIDA, as there is likely to be some variability across the ice crystal population since ice nucleation does not happen instantaneously (as can be seen in Figures 5-8, in examining how Si evolves). It is true that all crystal faces would not grow at the same rate, although since we unfortunately do not have direct experimental constraints on the ice crystal aspect ratios, we have made this assumption. We agree that this should be explored further, as we plan to do in our subsequent study on using Bayesian parameter estimation to put constraints on the parameters in the depositional ice growth model.

Figs. 5-8. It would be nice if the details in these plots were larger as everything is so small. I don't know the best way to do this except by adding more figures. I suppose T and P plots could be combined to create a little more space, and maybe the time axis could be trimmed to 1000 s, though these would only slightly help. On the other hand, I think it would be nice to see the inferred crystal sizes as well, so this would make things more cramped, perhaps suggesting the best solution is adding more figures. Also, in the plots, it would help to mark certain stages with a vertical line, such as the point when sublimation begins.

We have updated Figures 5-8 to only include the first 1000 s of each experiment, and combined the T and P plots. In the text, we have also clarified that the experiment time is relative to the

start of pumping (when cooling begins), that nucleation occurs up to the initial spike in supersaturation, and that the sublimation phase occurs when the temperature starts increasing (after the pressure change levels off).

Finally, I think the general style of figure captions is to put all the info needed to read the plots, but none of the interpretation. For example, in Fig. 6, the legends for exp. 40 "W2001" etc. should be defined in the caption.

For Fig. 6, we have added the abbreviations to the legend.

The interpretations (e.g., Fig. 5: "…also demonstrates very efficient ice growth…") belong in the main text.

This is a stylistic choice and we have chosen to leave it as is.

Line 360. Unclear about "…, as there is not yet consensus…supersaturation". Is this referring to the curve for layer nucleation rate vs supersaturation?

We are referring to the temperature dependence of the critical supersaturation (scrit) in the DisKICE model in these lines. There are multiple unknowns about how ice grows at low temperatures, not just uncertainty about the layer nucleation rates. In particular, it is still unclear how aerosol-ice interactions may play a role, and how these effects should be parameterized in models of depositional ice growth. We have clarified these lines to read:

"The functional form for the critical supersaturation ($s_{crit}$) in DiskICE at colder temperatures is not known, as there is not yet consensus about the theoretical basis for this critical supersaturation (Burton et al. 1951, Libbrecht 2005). The theoretical values are generally derived for the case of ice formation from pure water vapor, although chemical impurities and the existence of quasi-disordered layers could also significantly impact these values (Libbrecht 2005). Multiple unknowns still remain in the functional form of $s_{crit}$ for depositional ice growth in the atmosphere, including potential modifications required to account for the impact of nucleation pathway on ice formation."

Line 422. "mean absolute percentage error", or is this now a proper noun?
Fixed.

Line 494. Has "principal facet" been defined?

By "principal facet" we refer to the basal and prism facets of ice; so as not to define a new term in the conclusions we reworded this line (566-567) to read,

"More precise laboratory experiments are needed to constrain the distinct growth rates of the basal and prism facets at low temperatures, as the analysis presented here provides only an average growth rate (assuming isometric growth)."

**Reviewer 2**
The authors re-evaluate prior cloud chamber experiments on ice growth under cirrus cloud conditions. Their focus is to compare the experiments to model calculations based either on a constant deposition coefficient or with one, which depend on surface supersaturation. For experiments preformed above a temperature of 205 K, they find that both models are able to fit experimental data whereas at temperatures lower than 205 K the data indicate larger discrepancies to both models and on whether the nucleation occurred homogeneously or is heterogeneously induced.

The paper is well written, supported by adequate figures (sometimes a bit small to read), provides an overview over the state of the science for depositional ice growth in cirrus conditions and its topic well suited for the readership of ACP. I recommend publishing it, but ask the authors to take my comments into account when preparing a revised manuscript.

We thank the reviewer for their positive comments and for their careful reading of the manuscript. We have made the figures larger and reduced text in tables where possible so as to improve readability.

Of course, it is a bit disappointing that there is no clear difference between the two model approaches when comparing with experimental data as done in Fig. 9. The authors give an explanation in the last paragraph of section 5.4. They point out that there is only a limited crystal growth observed in the experiments, which may prohibit more distinct differences between the predictions of the modelling approaches. I feel the readers would profit a lot if the authors could compare the two approaches for a hypothetical experiment in which the differences become more apparent. Following this line of thought, could the authors even recommend an "updraft" speed and time span of observation, number density of ice crystals needed in an experiment to distinguish between a constant deposition coefficient and one depending on supersaturation?

This is a great point, and we plan to explore the question of whether AIDA expansion experiments are sensitive to the growth efficiency effect of the deposition coefficient in future work. The supersaturation-dependent deposition coefficient model is necessary for the growth of faceted ice crystals; as we have now pointed out in the Conclusions, simultaneously having observations of ice habits in the context of the AIDA experiments would provide an additional constraint on the deposition coefficient models.

In addition, I am not satisfied with the two sentences starting on line 464: "These results also demonstrate that the seeming contradiction between the high deposition coefficients previously observed in cloud chamber experiments (Skrotzki et al., 2013) and the low deposition coefficients observed in single particle levitation diffusion chamber experiments (Magee et al., 2006) can be resolved by a non-constant parameterization for $\alpha_D$. Here we have shown that both types of experiments can be consistently modeled with the same ice growth theory." I cannot see where the authors modeled the experiment of Magee et al. (2006), who came up

with a deposition coefficient of 6E-3. But I would welcome if the authors apply their supersaturation dependent deposition coefficient to the Magee et al. (2006) experimental data.

Modeling the results of Magee et al. 2006 with a non-constant parameterization was previously done in Harrington et al. (2019, J Atmos Sci). That study also showed that Magee's results are consistent with the growth data from Bailey and Hallett (2004, 2010). While a variable deposition coefficient can explain these data, it is also important to keep the limitations of this theory in mind. As Harrington et al. (2019) point out, using a single variable (deposition coefficient) to characterize all of the surface processes that can influence growth is a rough approximation. Other processes certainly affect the growth of newly formed ice crystals. For instance, prior data show that after crystal formation, facets develop on the crystal surface and this produces growth along the surface (Gonda and Yamazaki, 1984; Nelson and Swanson, 2019). Pokrifka et al. (2020) show that facet development can explain the relatively constant growth rates that are sometimes measured in growth chambers. Harrington and Pokrifka (2021) show that facet development after ice nucleation, or during facet regrowth after sublimation, can produce relatively constant growth rates. These constant rates could be mis-interpreted as a surface kinetic limitation, when they are instead due to a different mode of growth. All prior growth measurements likely convolve facet development with surface kinetics, and until there is a satisfactory method to measure this growth and model it, this limitation should be borne in mind. It is worth noting that this is one of many limitations to the use of a variable deposition coefficient to model crystal surface processes. To clarify that we were referring both to this previous study (in regards to Magee et al. 2006) and to the current work (in regards to Skrotzki et al. 2013), and to emphasize the limitations to these interpretations, we have reworded these lines (503-515) to read,

"Harrington et al (2019) demonstrated that the low deposition coefficients observed in single particle levitation diffusion chamber experiments (Magee et al., 2006) are consistent with the temperature, super-saturation, and facet-dependent parameterization of $\alpha_{D}$ given by Eq. 3. Here we have shown that the high deposition coefficients previously observed in cloud chamber experiments (Skrotzki et al., 2013) can also be explained by this same ice growth theory. Thus, the seeming contradiction between levitation diffusion chamber experiments and cloud chamber studies can be resolved by a non-constant parameterization for $\alpha_{D}$. However, caution is warranted. While constant deposition coefficients cannot be correct for faceted ice, other processes can affect the growth rates. For instance, immediately following nucleation, facets develop on frozen droplets and grow along the surface. This growth can produce relatively constant mass growth rates (Pokrifka et al., 2020) that can be mis-interpreted as a strong kinetic limitation (Harrington and Pokrifka, 2021). This is also true for facet regrowth after sublimation, which can lead to much weaker growth (Harrington and Pokrifka, 2021). Moreover, whether ice forms from heterogeneous nuclei or homogeneous freezing may impact the growth rate (Pokrifka et al., 2020). Since our current theories of ice crystal growth are relatively simple, all surface processes are convolved with the deposition coefficients, and this limitation should be borne in mind when using any deposition coefficient model (Harrington et al., 2019)."

A last comment: In all figures showing the comparison of model with experimental data you take the input data for the model (temperature, pressure and ice particle concentration) as exact. A few sensitivity calculations (e.g. varying ice number density by 5 %) would be quite helpful. Or phrased differently how sensitive is the model to these constraints?

We are exploring this issue more thoroughly in the second part of this research study, using a Markov Chain Monte Carlo algorithm to use Bayesian inference to place constraints on the parameters in the depositional ice growth model. Systematic inference to improve the representation of ice microphysical processes can take the form of adjusting structural assumptions of models of particle evolution, or the parameters within these models– in this first part we have chosen to focus on the first issue.

Technical comments:
Line 25: I find that "surface complexity" needs a bit more of explanation at this point in the introduction.

We have replaced the term "surface complexity" with "ice crystal complexity", and added this explanation to the text in lines 25 - 30:

"In this study, ice crystal complexity refers to surface distortions that affect single ice crystals, in terms of their surface roughness at different scales, polycrystallinity, and hollowing (Schnaiter et al. 2016)."

Section 2.1: You partly review the literature in the section. For the reader it may prove helpful if you add the temperatures of these studies, so that the reader does not need to look those up.

While a full review of the literature is beyond the scope of this work, Harrington et al. 2019 did a more thorough review of past experimental constraints for $s_{crit}$, and we have added a more clear reference to this in Section 2.1 in lines 166-167:

"Harrington et al. 2019 gives a more full review of past experimental constraints on $s_{crit}$ and their relevant temperature ranges."

Line 279: I find the term "vapor tendency" very odd, but I am not a native speaker.

By this we mean the instantaneous change in vapor, which is typical terminology used in the literature, so we have chosen to leave this as is.

Line 284: again English: I suggest to delete the "however"
Updated.

Figure caption Fig.9.: there is no panel "g", it should read panel "f".
Fixed.

References:

Gonda, T. and Yamazaki, T., 1984. Initial growth forms of snow crystals growing from frozen cloud droplets. *Journal of the Meteorological Society of Japan. Ser. II*, *62*(1), pp.190-192.

Harrison, A., Moyle, A. M., Hanson, M., and Harrington, J. Y.: Levitation Diffusion Chamber Measurements of the Mass Growth of Small Ice Crystals from Vapor, Journal of the Atmospheric Sciences, 73, 2743–2758, DOI:10.1175/JAS–D–15–0234.1, 2016.

Harrington, J. Y., Moyle, A., Hanson, L. E., and Morrison, H.: On Calculating Deposition Coefficients and Aspect-Ratio Evolution in Ap- proximate Models of Ice Crystal Vapor Growth, Journal of the Atmospheric Sciences, 76, 1609–1625, DOI:10.1175/JAS–D–18–0319.1, https://doi.org/10.1175/JAS-D-18-0319.1, 2019.

Pokrifka, G.F., Moyle, A.M., Hanson, L.E. and Harrington, J.Y., 2020. Estimating surface attachment kinetic and growth transition influences on vapor-grown ice crystals. *Journal of the Atmospheric Sciences*, *77*(7), pp.2393-2410.

Sarkozy, L.C., Clouser, B.W., Lamb, K.D., Stutz, E.J., Saathoff, H., Möhler, O., Ebert, V. and Moyer, E.J., 2020. The Chicago Water Isotope Spectrometer (ChiWIS-lab): A tunable diode laser spectrometer for chamber-based measurements of water vapor isotopic evolution during cirrus formation. *Review of Scientific Instruments*, *91*(4), p.045120.

Schnaiter, M., Järvinen, E., Vochezer, P., Abdelmonem, A., Wagner, R., Jourdan, O., Mioche, G., Shcherbakov, V. N., Schmitt, C. G., Tricoli, U., et al.: Cloud chamber experiments on the origin of ice crystal complexity in cirrus clouds, Atmospheric Chemistry and Physics, 16, 5091–5110, 2016.

Zhang, C. and Harrington, J. Y.: Including surface kinetic effects in simple models of ice vapor diffusion, Journal of the Atmospheric Sciences, 71, 372–390, 2014.

Zhang, C. and Harrington, J. Y.: The effects of surface kinetics on crystal growth and homogeneous freezing in parcel simulations of cirrus, Journal of the Atmospheric Sciences, 72, 2929–2946, 2015.